# Scale-Aware Pretraining of Time Series Foundation Models via Multi-Patch Token Alignment and Hybrid Masking

## Abstract

Pretraining time series foundation models across diverse datasets necessitates effective handling of varying sampling frequencies. A prevalent approach assigns dataset-specific patch sizes based on sampling rates and employs separate MLPs for token projection, which leads to fragmented representations across scales and hinders alignment and transferability. In contrast, some studies enforce a fixed patch size across datasets to ensure consistency, yet this uniformity neglects inherent temporal variations and often causes information loss. To address these challenges, we propose a scale-aware token alignment mechanism that treats the patch size used during input segmentation as an explicit notion of scale. By incorporating contrastive learning across scales, our approach aligns the representation spaces induced by different MLPs while preserving their distinct modeling capacities. On top of this aligned representation, we introduce a hybrid masking strategy that enables multi-scale temporal understanding at the token level. By combining random and contiguous masking, the model learns to recover both fine-grained patterns and long-range temporal structures during pretraining. Experiments on benchmark datasets show that our approach consistently improves forecasting performance, highlighting the benefits of scale-aware token alignment and multi-scale understanding in time series model pretraining.

## 1 Introduction

The recent emergence of foundation models has significantly advanced various domains such as natural language processing (Brown et al., 2020; Dubey et al., 2024), computer vision (Oquab et al., 2023; Radford et al., 2021), and speech understanding (Baevski et al., 2020; Radford et al., 2023). Inspired by their success, growing efforts have been devoted to developing foundation models for time series, aiming to produce general-purpose representations transferable across diverse downstream tasks. An early line of work adapts pretrained language models to time series tasks, leveraging their sequence modeling capabilities in hopes of achieving strong generalization (Cao et al., 2023; Jin et al., 2023; Pan et al., 2024). However, the modality gap often hinders their performance on temporally structured data, resulting in suboptimal generalization across diverse time series tasks. Moreover, their black-box nature further exacerbates the issue, raising concerns about interpretability and the lack of alignment with intrinsic temporal characteristics (Tan et al., 2024). To address these challenges, a second line of work has emerged that trains foundation models from scratch on large-scale, heterogeneous time series datasets (Shi et al., 2024; Woo et al., 2024; Ansari et al., 2024). These models aim to capture universal temporal dynamics in a data-driven and domain-adaptive manner, thereby enhancing robustness to distribution shifts and improving transferability across domains with varying sampling rates, modalities, and sequence lengths (e.g., finance, healthcare, meteorology, IoT).

Despite the promise of the latter direction, it presents unique challenges—particularly in how to effectively segment and tokenize continuous signals for cross-dataset pretraining. Unlike language, where discrete word units naturally serve as stable tokens (Sennrich et al., 2016), or vision, where uniform patch sizes are viable due to consistent spatial resolution and semantic robustness (Touvron et al., 2021; Dosovitskiy et al., 2020), time series data exhibit irregular sampling and variable se-

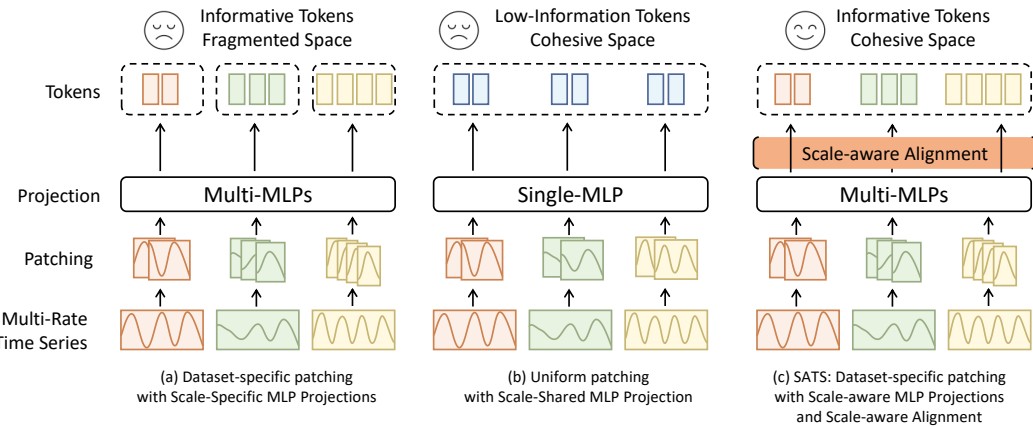

Figure 1: (a) Dataset-specific patch sizes and independent MLPs for varying sampling rates lead to fragmented token spaces. (b) Using a unified patch size and MLP risks information bottlenecks and misaligned local dynamics. (c) SATS adopts dataset-specific patch sizes and enforces scale-aware alignment across MLP-projected spaces, yielding semantically rich and consistent representations.

quence lengths, making fixed-size downsampling ineffective. These characteristics necessitate the use of small, adaptive patch sizes to preserve fine-grained temporal patterns.

As shown in Figure 1, recent studies have explored two main strategies for time series tokenization, each with inherent limitations. (1) **Dataset-specific patching** adopts variable patch sizes tailored to local sampling rates, combined with independent MLPs for token projection (Zhang et al., 2024; Woo et al., 2024). While this design aligns well with the granularity of each dataset, it results in fragmented token spaces that hinder the learning of generalizable temporal patterns and compromise training stability. (2) **Uniform patching** enforces a globally small patch size across datasets to promote representational consistency (Wang et al., 2025; Liu et al., 2024b). However, this strategy introduces information bottlenecks and often misaligns local dynamics, as it fails to accommodate the diverse temporal structures inherent in different datasets. Both strategies, therefore, face a trade-off between dataset adaptability and representational generality, limiting their effectiveness in scalable pretraining.

To bridge the gap between fragmented token spaces introduced by adaptive patching and the representational rigidity of fixed segmentation, we propose a scale-aware token alignment mechanism tailored for time series pretraining. By treating the patch size as an explicit notion of scale, our method aligns the representation spaces induced by scale-specific MLPs. This is achieved by minimizing the distance between mean token embeddings across scales to encourage semantic alignment, while simultaneously maximizing the distance between their maximal embeddings to preserve the scale-specific modeling capacity. The resulting token space offers a unified yet expressive foundation for downstream tasks.

Building on this aligned representation space, a remaining challenge lies in the diverse temporal structures inherent to different datasets. Even with aligned embeddings, temporal variations may manifest within individual tokens or span across multiple tokens, depending on the dynamics of the underlying sequence. To capture such variability, we introduce a hybrid masking strategy that enhances multi-scale temporal modeling during masked reconstruction. This strategy combines random masking, which promotes fine-grained inference, with contiguous masking, which facilitates the modeling of long-range dependencies. By jointly optimizing across these complementary patterns, the model learns to recover temporal structures at varying resolutions, improving its robustness and generalization.

Our main contributions are summarized as follows:

- We propose **SATS**, a **S**cale-**A**ware foundation model for **T**ime **S**eries, which achieves superior generalization across diverse datasets.
- We introduce a scale-aware alignment mechanism based on scale-specific MLPs, unifying token spaces across patch scales while preserving scale-specific expressiveness.

- We design a hybrid masking strategy that enables the model to capture both fine-grained and long-range temporal dependencies across multiple resolutions.

- Extensive experiments demonstrate the effectiveness of SATS in both zero-shot and in-distribution forecasting settings, establishing its potential as a strong pretraining paradigm for time series foundation models.

## 2 RELATED WORK

**Time Series Foundation Models**   Large language models (LLMs) have recently been introduced into time series forecasting through prompt tuning or direct fine-tuning (Pan et al., 2024; Cao et al., 2023; Zhou et al., 2023). While these methods leverage pretrained knowledge, they often face challenges such as domain mismatch, limited token expressiveness, and modality entanglement (Liu et al., 2024a; Jin et al., 2023). These issues not only hinder effective representation learning but also obscure the mechanisms by which LLMs capture temporal dependencies (Tan et al., 2024). Moreover, their reliance on dataset-specific training limits robustness under distribution shifts, prompting increasing interest in pretraining-based time series models.

In response, a new line of research has focused on pretraining time series foundation models natively on large-scale temporal data, aiming to learn general-purpose representations without relying on language-centric priors or external modalities. Owing to the inherent characteristics of forecasting tasks—such as unidirectional temporal dependency, variable-length prediction horizons, and strong autoregressive inductive biases—decoder-based architectures have garnered increasing attention. For instance, decoder-only models such as Timer (Liu et al., 2024c) and Lag-Llama (Rasul et al., 2023) adopt causal architectures tailored for forecasting, with the latter incorporating lagged covariates for improved accuracy. Sparse MoE variants like Time-MoE (Shi et al., 2024) and Moirai-MoE (Liu et al., 2024b) further enhance scalability. In contrast, encoder-decoder models like Light-GTS (Wang et al., 2025) and Chronos (Ansari et al., 2024) leverage parallel decoding and discretized training objectives to capture temporal patterns. In contrast, encoder-only architectures remain a relatively underexplored branch in the context of time series foundation models. The design of effective pretraining tasks for such models is still unsettled (Woo et al., 2024; Goswami et al., 2024). Notably, recent theoretical analyses (Yao et al., 2024) suggest that encoder-only models exhibit higher power-law scaling exponents, indicating stronger representational capacity under limited compute. These findings highlight the untapped potential of encoder-only backbones, motivating further investigation into their architecture and pretraining strategies in the temporal domain.

**Contrastive Learning in Pretraining**   Contrastive learning has emerged as a powerful paradigm in large-scale pretraining across various domains. In NLP, methods such as SimCSE (Gao et al., 2021) leverage contrastive objectives to learn semantically meaningful sentence embeddings without supervision. In computer vision, CLIP (Radford et al., 2021) and ALIGN (Jia et al., 2021) jointly embed images and texts by maximizing the similarity of paired modalities while contrasting unpaired ones, achieving impressive zero-shot performance. While contrastive learning in time series remains relatively underexplored, recent works like TS-TCC (Eldele et al., 2021) and CoST (Woo et al., 2022) demonstrate its potential in learning transferable representations by aligning augmented views of temporal data. A key advantage of contrastive learning lies in its ability to preserve embedding diversity—by pulling semantically similar instances closer and pushing dissimilar ones apart, it structures the latent space in a discriminative and robust manner. Inspired by contrastive learning's structured divergence, we adopt an InfoNCE-motivated objective to enhance distinctiveness among multi-scale features—without explicit negative samples—thus inheriting its regularization benefits.

## 3 METHODOLOGY

**Problem Formulation**   Let $\mathcal{S} = \{(\mathbf{X}^{(i)}, \mathbf{C}^{(i)})\}_{i=1}^{N}$ denote a dataset of multivariate time series, where $\mathbf{X}^{(i)} \in \mathbb{R}^{d_x \times T_i}$ are target sequences and $\mathbf{C}^{(i)} \in \mathbb{R}^{d_c \times T_i}$ are associated covariates. Given the unmasked observations $\mathbf{X}_{\text{obs}}$ and the corresponding covariates $\mathbf{C}$, the objective is to learn model parameters $\theta$ such that the model $f_\theta$ predicts the distribution parameters $\hat{\psi}$ for the masked subset $\mathbf{X}_{\mathcal{M}}$ of the target sequence.

This leads to the following optimization problem:

$$\min_{\theta} \; \mathbb{E}_{(\mathbf{X},\mathbf{C})\sim p(\mathcal{S})} \, \mathbb{E}_{\mathcal{M}\sim p(\mathcal{T}|\mathcal{S})} \left[ \mathcal{L}_{\text{nll}}\left( \mathbf{X}_{\mathcal{M}}, \hat{\psi} \right) \right] \quad \text{s.t.} \quad \hat{\psi} = f_{\theta}(\mathbf{X}_{\text{obs}}, \mathbf{C}) \tag{1}$$

Here, $\mathcal{L}_{\text{nll}}$ denotes the *negative log-likelihood loss*:

$$\mathcal{L}_{\text{nll}}(\mathbf{X}_{\mathcal{M}}, \hat{\psi}) = -\log p(\mathbf{X}_{\mathcal{M}} \mid \hat{\psi}) \tag{2}$$

where $p(\mathcal{S})$ is the data-generating distribution over time series instances $(\mathbf{X}, \mathbf{C})$, and $p(\mathcal{T} \mid \mathcal{S})$ defines the task sampling distribution that governs the selection of masked positions $\mathcal{M} \subset \{1, \dots, T\}$ for prediction. Classical forecasting corresponds to the special case where the masked region $\mathcal{M}$ is located at the end of the sequence.

## 3.1 MODEL ARCHITECTURE

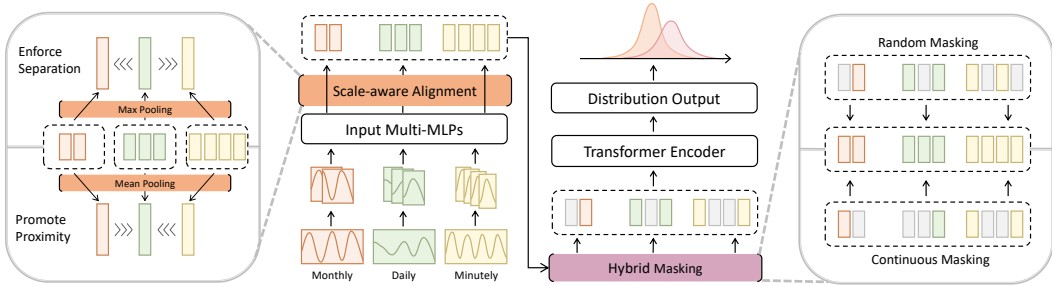

Figure 2: Overview of the SATS framework. Tokens from multiple patch sizes are projected via separate MLPs. SATS employs **Scale-aware Alignment** mechanism to promote proximity of mean-pooled representations within each scale, while enforcing separation of max-pooled representations across scales—balancing consistency and scale-specific expressiveness. **Hybrid masking strategy**, integrating Random Masking and Continuous Masking, is further applied to capture both fine-grained and long-range temporal dependencies.

As shown in Figure 2, SATS adopts a non-overlapping patch-based, encoder-only Transformer (Nie et al., 2022). The multivariate time series is first flattened and, following Moirai (Woo et al., 2024), mapped into patches of varying sizes based on the dataset. To improve efficiency, we adopt packing as a default setting (Krell et al., 2021; Dubey et al., 2024), enabling tokens with different patch sizes from multiple datasets to be packed into a single sequence. This multi-scale design introduces inconsistencies in the token space; while packing is not the direct cause, it is an indispensable component of modern scalable training, making it both practical and necessary to develop solutions within this paradigm.

To mitigate such inconsistencies while embracing the packing paradigm, SATS employs a scale-aware alignment mechanism: it pulls closer the mean-pooled representations within the same scale, while pushing apart the max-pooled ones across scales, ensuring consistency while preserving scale-specific expressiveness. Based on this aligned space, a hybrid masking strategy combining random and contiguous patterns is applied to capture both fine-grained and long-range dependencies.

Although not shown, the encoder incorporates key techniques from foundational model pretraining—such as RoPE (Su et al., 2021), SwiGLU (Shazeer, 2020), and RMSNorm (Zhang & Sennrich, 2019)—as well as inductive biases specific to time-series pretraining, including Any-Variate Bias, Mixture Distribution Output (Woo et al., 2024) and RevIN (Kim et al., 2021) for modeling inter-variable dependencies and normalization under distribution shifts.

**Scale-aware Alignment** To enhance the effectiveness of temporal modeling, especially when dealing with subsequences of varying scales, it is crucial to design an effective alignment strategy. Given token sequences $\mathcal{I} \in \mathbb{R}^{L \times D}$, where $L$ represents the maximum input length during training and $D$ is the hidden layer dimension of the encoder, the challenge arises from the coexistence of tokens originating from $n \leq N$ different patch sizes, where $N$ denotes the total number of distinct patch sizes. A direct approach could be to minimize the feature space distance, such as

cosine similarity, between subsequences, encouraging their proximity. However, this approach faces several challenges: first, the varying lengths of subsequences make it difficult to quantify alignment; second, different samples within the same batch may contain different numbers of subsequences, complicating the application of proximity constraints both within and across samples. Furthermore, to avoid feature collapse, a structured information constraint is necessary, as it prevents the model from mapping features into a low-rank subspace, thus maintaining the richness of temporal representations.

In response to these challenges, we propose the Scale-aware Alignment method, which integrates two key components. First, we introduce a pooling mechanism to address the issues of variable subsequence lengths and differing numbers of subsequences across samples. Specifically, we pool the samples based on their patch sizes to generate the embedding representation $Y \in \mathbb{R}^{N \times D}$. In cases where a patch size is absent in a given sample, the corresponding embedding position $Y_i$ is set to zero ($i \leq N$), thereby preventing gradient propagation from this missing patch. Second, inspired by the principles of contrastive learning, we design a structured information constraint: the mean embeddings from different patch sizes are pulled closer to establish neighboring centers in the token space, while the maximal embeddings are repelled to encode scale-specific information, ensuring richer and more diverse token semantics. More theoretical analysis is provided in Appendix A. To operationalize this constraint, we adopt the InfoNCE framework, as detailed in Equation 3 and Equation 4, where $\cos(\cdot)$ denotes the cosine similarity function and $\tau$ is the temperature parameter.

$$\mathcal{L}_{\text{close}} = -\mathbb{E}\left[\log\left(\frac{\sum_{j \neq i} \exp\left(\cos(Y_i \cdot Y_j)/\tau\right)}{\sum_{j=1}^{N} \exp\left(\cos(Y_i \cdot Y_j)/\tau\right)}\right)\right] \tag{3}$$

$$\mathcal{L}_{\text{far}} = -\mathbb{E}\left[1 - \log\left(\sum_{j=1}^{N} \exp\left(\cos(Y_i \cdot Y_j)/\tau\right)\right)\right] \tag{4}$$

In practice, $Y_i \in Y^{\text{mean}}$ is sequentially substituted into Equation 3, while $Y_i \in Y^{\text{max}}$ is substituted into Equation 4. Although both equations follow the InfoNCE form, they do not involve true negative samples. We therefore combine these two losses to form the final scale-aware alignment constraint in Equation 5. This design provides structured regularization that aligns feature representations across different patch sizes, enhancing cross-scale consistency while preventing representation collapse. The hyperparameter $\beta$ controls the relative weight of the maximal embedding pull-away term, balancing the overall objective.

$$\mathcal{L}_{\text{saa}} = \mathcal{L}_{\text{close}} + \beta \mathcal{L}_{\text{far}} \tag{5}$$

**Hybrid Masking Strategy**  On top of the aligned token space, the intrinsic heterogeneity and complexity of temporal dynamics across datasets continue to challenge effective representation learning. Although alignment mitigates certain variations, temporal dependencies inherently span multiple scales: some manifest as fine-grained, localized fluctuations within individual tokens, while others emerge as extended, structured patterns across contiguous token segments. To comprehensively capture these diverse temporal scales and improve the robustness of learned representations, we therefore propose a hybrid masking strategy that synergistically combines random masking with contiguous masking during pretraining.

Concretely, given each token subsequence $\mathcal{I}_j \in \mathbb{R}^{L_j \times D}$ extracted from the full sequence $\mathcal{I}$, where $L_j$ denotes the length of the $j$-th subsequence, a masking ratio $r \in [0.15, 0.5]$ is applied. For each subsequence, a predefined probability $p \in [0, 1]$ determines whether random or contiguous masking is used. With probability $p$, random masking uniformly selects $m_j$ token positions, where $m_j = \lceil r \cdot L_j \rceil$, producing a binary mask $\mathcal{M}_r^{(j)}$:

$$\mathcal{M}_r^{(j)}(i) = \begin{cases} 1, & \text{if token } i \text{ is randomly selected} \\ 0, & \text{otherwise} \end{cases} \quad \text{s.t. } \sum_{i=0}^{L_j-1} \mathcal{M}_r^{(j)}(i) = m_j. \tag{6}$$

Alternatively, with probability $1 - p$, contiguous masking is applied by sampling a start index $s_j \in \{0, \ldots, L_j - m_j\}$, masking a continuous block of tokens:

$$\mathcal{M}_c^{(j)}(i) = \begin{cases} 1, & s_j \leq i < s_j + m_j \\ 0, & \text{otherwise.} \end{cases} \tag{7}$$

The final mask $\mathcal{M}^{(j)}$ applied to each subsequence $\mathcal{I}_j$ is sampled as

$$\mathcal{M}^{(j)} = \begin{cases} \mathcal{M}_r^{(j)}, & \text{with probability } p \\ \mathcal{M}_c^{(j)}, & \text{with probability } 1 - p. \end{cases} \quad (8)$$

By guiding the model to recover masked tokens across both randomly distributed and contiguous spans, this probabilistic hybrid masking balances fine-grained local inference and long-range dependency learning. Consequently, it enhances the robustness and generalizability of learned representations for diverse temporal modeling tasks.

## 3.2 MODEL TRAINING

**Unified Learning Objective**   Both the Scale-aware Alignment and the Hybrid Masking Strategy are parameter-free, which not only simplifies their integration but also allows them to be seamlessly combined into a unified learning objective without introducing additional model complexity. In practice, the mask $\mathcal{M}$ obtained from Equation 8 is applied to Equation 2 to compute the primary training loss. Simultaneously, Equation 5 is employed as an auxiliary training loss to enforce the Scale-aware Alignment. We combine them into the total loss function as follows:

$$\mathcal{L} = \mathcal{L}_{\text{nll}} + \alpha \mathcal{L}_{\text{saa}} \quad (9)$$

where $\alpha$ is a weighting coefficient balancing the two objectives.

**SATS Setup**   We pretrain the SATS models on the LOTSA dataset (Woo et al., 2024) in two configurations—small and base—with detailed model specifications provided in Table 1. The small model is trained for 100,000 steps with a batch size

Table 1: Key parameter details of SATS model sizes.

|       | Layers | $d_{\text{model}}$ | $d_{\text{ff}}$ | Heads | Params |
|-------|--------|--------|--------|-------|--------|
| SATS$_S$ | 6 | 384 | 1536 | 6 | 14M |
| SATS$_B$ | 9 | 768 | 3072 | 12 | 70M |

of 128, while the base model is trained for 200,000 steps with a batch size of 64. Both configurations employ the AdamW optimizer and follow a learning rate schedule consisting of 10,000 linear warmup steps followed by cosine annealing. The initial learning rate is set to 1e-3 and the weight decay to 1e-1. Further details on hyperparameters and implementation are provided in Appendix B.

## 4 EXPERIMENTS

### 4.1 BENCHMARKING SETUP

**Baselines**   We conduct extensive comparisons with widely adopted foundation models for time series, including Timer-XL (Liu et al., 2025), Time-MoE (Shi et al., 2024), Moirai (Woo et al., 2024), Chronos (Ansari et al., 2024), Moment (Goswami et al., 2024), TimesFM (Das et al., 2024) and LLMTime (Gruver et al., 2024). In response to Bergmeir, we further expand our evaluation under the in-distribution setting by incorporating a broader range of baselines, including classical methods such as Naive, ETS (Hyndman et al., 2008), and DeepAR (Salinas et al., 2019).

**Evaluation Setup**   To ensure a fair comparison, all baselines are implemented following their original settings as reported in the respective papers to reproduce their best performance. Following Moirai (Woo et al., 2024), we configure SATS by selecting context lengths from $\{1000, 2000, 3000, 4000, 5000\}$ and determining patch sizes based on frequency. Detailed evaluation protocols and error bars are provided in Appendix B.4.

### 4.2 ZERO-SHOT FORECASTING

**Setup**   We start by conducting out-of-distribution evaluations on five widely-used benchmark datasets that are not included in LOTSA. Following standard practice, we consider four prediction horizons $\{96, 192, 336, 720\}$ and adopt MSE and MAE as evaluation metrics. To ensure fair comparison, for models with multiple variants, we exclude those with more than 1B parameters and report results from the variant with the best average performance.

Table 2: Full results of zero-shot forecasting across all evaluated models. Lower values of MSE and MAE indicate superior performance. As TimesFM incorporates Weather data during pretraining, it is excluded from evaluation on this dataset (denoted by "–"). **Red** highlights the best result, while Blue marks the second best. More results and the rationale for dataset selection can be found in Appendix C.1.

| Models | $\text{SATS}_\text{S}$ | | $\text{SATS}_\text{B}$ | | Timer-XL | | $\text{Time-MoE}_\text{B}$ | | $\text{Moirai}_\text{B}$ | | $\text{Chronos}_\text{L}$ | | Moment | | TimesFM | |
|---|---|---|---|---|---|---|---|---|---|---|---|---|---|---|---|---|
| Metrics | MSE | MAE | MSE | MAE | MSE | MAE | MSE | MAE | MSE | MAE | MSE | MAE | MSE | MAE | MSE | MAE |
| ETTh1 96 | 0.375 | 0.393 | 0.360 | 0.387 | 0.369 | 0.391 | 0.357 | 0.381 | 0.383 | 0.402 | 0.441 | 0.390 | 0.688 | 0.557 | 0.414 | 0.404 |
| ETTh1 192 | 0.412 | 0.415 | 0.395 | 0.409 | 0.405 | 0.413 | 0.384 | 0.404 | 0.425 | 0.429 | 0.502 | 0.424 | 0.688 | 0.560 | 0.465 | 0.434 |
| ETTh1 336 | 0.423 | 0.425 | 0.413 | 0.422 | 0.418 | 0.423 | 0.411 | 0.434 | 0.456 | 0.450 | 0.576 | 0.467 | 0.675 | 0.563 | 0.503 | 0.456 |
| ETTh1 720 | 0.418 | 0.441 | 0.413 | 0.438 | 0.423 | 0.441 | 0.449 | 0.477 | 0.470 | 0.473 | 0.835 | 0.583 | 0.683 | 0.585 | 0.511 | 0.481 |
| ETTh1 AVG | 0.407 | 0.418 | 0.395 | 0.414 | 0.404 | 0.417 | 0.400 | 0.424 | 0.433 | 0.438 | 0.589 | 0.466 | 0.684 | 0.566 | 0.473 | 0.444 |
| ETTh2 96 | 0.283 | 0.328 | 0.273 | 0.331 | 0.283 | 0.342 | 0.305 | 0.359 | 0.277 | 0.327 | 0.320 | 0.345 | 0.342 | 0.396 | 0.315 | 0.349 |
| ETTh2 192 | 0.343 | 0.369 | 0.330 | 0.372 | 0.340 | 0.379 | 0.351 | 0.386 | 0.340 | 0.374 | 0.406 | 0.399 | 0.388 | 0.402 | 0.388 | 0.395 |
| ETTh2 336 | 0.365 | 0.391 | 0.353 | 0.396 | 0.366 | 0.400 | 0.391 | 0.418 | 0.371 | 0.401 | 0.492 | 0.453 | 0.356 | 0.407 | 0.422 | 0.427 |
| ETTh2 720 | 0.404 | 0.424 | 0.380 | 0.409 | 0.397 | 0.431 | 0.419 | 0.454 | 0.394 | 0.426 | 0.603 | 0.511 | 0.395 | 0.434 | 0.443 | 0.454 |
| ETTh2 AVG | 0.349 | 0.378 | 0.334 | 0.388 | 0.347 | 0.388 | 0.367 | 0.404 | 0.345 | 0.382 | 0.455 | 0.427 | 0.362 | 0.410 | 0.392 | 0.406 |
| ETTm1 96 | 0.325 | 0.353 | 0.323 | 0.345 | 0.317 | 0.356 | 0.338 | 0.368 | 0.396 | 0.382 | 0.457 | 0.403 | 0.654 | 0.527 | 0.361 | 0.370 |
| ETTm1 192 | 0.352 | 0.372 | 0.352 | 0.364 | 0.358 | 0.381 | 0.353 | 0.388 | 0.425 | 0.402 | 0.530 | 0.450 | 0.662 | 0.532 | 0.414 | 0.405 |
| ETTm1 336 | 0.372 | 0.387 | 0.371 | 0.379 | 0.386 | 0.401 | 0.381 | 0.413 | 0.452 | 0.415 | 0.577 | 0.481 | 0.672 | 0.537 | 0.445 | 0.429 |
| ETTm1 720 | 0.405 | 0.410 | 0.401 | 0.403 | 0.430 | 0.431 | 0.504 | 0.493 | 0.477 | 0.431 | 0.660 | 0.526 | 0.692 | 0.551 | 0.512 | 0.471 |
| ETTm1 AVG | 0.364 | 0.380 | 0.362 | 0.373 | 0.373 | 0.392 | 0.394 | 0.416 | 0.437 | 0.407 | 0.556 | 0.465 | 0.670 | 0.537 | 0.433 | 0.419 |
| ETTm2 96 | 0.172 | 0.255 | 0.167 | 0.251 | 0.189 | 0.277 | 0.201 | 0.291 | 0.195 | 0.269 | 0.197 | 0.271 | 0.260 | 0.335 | 0.202 | 0.270 |
| ETTm2 192 | 0.226 | 0.292 | 0.222 | 0.290 | 0.241 | 0.315 | 0.258 | 0.334 | 0.247 | 0.303 | 0.254 | 0.314 | 0.289 | 0.350 | 0.289 | 0.321 |
| ETTm2 336 | 0.279 | 0.327 | 0.269 | 0.323 | 0.286 | 0.348 | 0.324 | 0.373 | 0.291 | 0.333 | 0.313 | 0.353 | 0.324 | 0.369 | 0.360 | 0.366 |
| ETTm2 720 | 0.369 | 0.385 | 0.343 | 0.374 | 0.375 | 0.402 | 0.488 | 0.464 | 0.355 | 0.377 | 0.416 | 0.415 | 0.394 | 0.409 | 0.462 | 0.430 |
| ETTm2 AVG | 0.262 | 0.315 | 0.250 | 0.309 | 0.273 | 0.336 | 0.318 | 0.366 | 0.272 | 0.321 | 0.295 | 0.338 | 0.317 | 0.366 | 0.328 | 0.347 |
| Weather 96 | 0.180 | 0.236 | 0.162 | 0.217 | 0.171 | 0.225 | 0.160 | 0.214 | 0.176 | 0.210 | 0.194 | 0.235 | 0.243 | 0.255 | - | - |
| Weather 192 | 0.226 | 0.280 | 0.210 | 0.265 | 0.221 | 0.271 | 0.210 | 0.260 | 0.218 | 0.251 | 0.249 | 0.285 | 0.278 | 0.329 | - | - |
| Weather 336 | 0.274 | 0.316 | 0.258 | 0.302 | 0.274 | 0.311 | 0.274 | 0.309 | 0.267 | 0.288 | 0.302 | 0.327 | 0.306 | 0.346 | - | - |
| Weather 720 | 0.341 | 0.363 | 0.325 | 0.349 | 0.356 | 0.370 | 0.418 | 0.405 | 0.338 | 0.338 | 0.372 | 0.378 | 0.350 | 0.374 | - | - |
| Weather AVG | 0.255 | 0.299 | 0.239 | 0.283 | 0.256 | 0.294 | 0.266 | 0.297 | 0.250 | 0.271 | 0.279 | 0.306 | 0.294 | 0.326 | - | - |
| Average | 0.327 | 0.358 | 0.316 | 0.351 | 0.330 | 0.365 | 0.349 | 0.381 | 0.348 | 0.364 | 0.435 | 0.401 | 0.465 | 0.441 | - | - |
| $1^{st}$ Count | 4 | | 36 | | 1 | | 6 | | 5 | | 0 | | 0 | | 0 | |

**Result** The detailed zero-shot results are presented in Table 2, where $\text{SATS}_\text{B}$ consistently achieves state-of-the-art performance. Compared to $\text{Moirai}_\text{B}$, the strongest encoder-only baseline, $\text{SATS}_\text{B}$ achieves a **9.2%** improvement in MSE. It also outperforms Timer-XL (decoder-only) and $\text{Chronos}_\text{L}$ (encoder-decoder) with MSE improvements of **4.2%** and **27.4%**, respectively. Notably, $\text{SATS}_\text{B}$ contains only 70M parameters, which is substantially fewer than those of the compared baselines. Moreover, even the lightweight $\text{SATS}_\text{S}$ with 14M parameters surpasses all other baselines in overall average performance, highlighting its efficiency.

## 4.3 IN-DISTRIBUTION FORECASTING

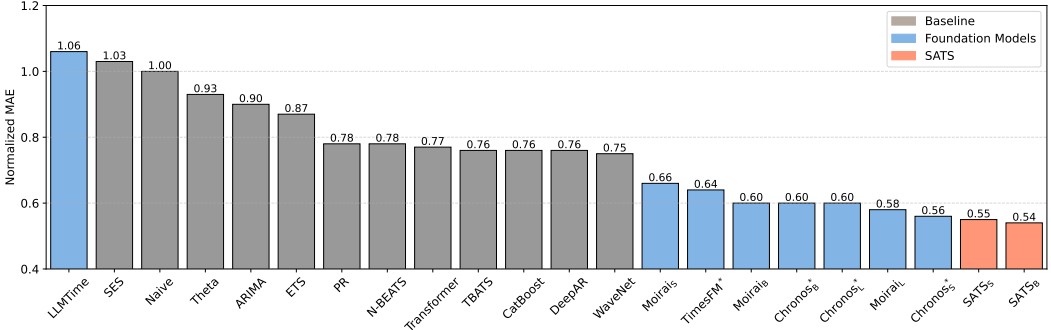

Figure 3: In-distribution forecasting performance evaluated on 29 datasets from the Monash benchmark (Godahewa et al., 2021). Methods trained with access to these evaluation datasets during pretraining are denoted with asterisks (*). Results are normalized using the naive forecast and summarized with the geometric mean. The detailed results are listed in Appendix C.2.

**Setup** We conduct an in-distribution evaluation on 29 datasets sourced from the Monash benchmark (Godahewa et al., 2021), where only the training portions are included in LOTSA and the test sets are reserved for evaluation. We report the normalized MAE, calculated by dividing each model's

MAE by that of a naive forecast, and aggregate the results using the geometric mean across datasets, providing a concise yet comprehensive assessment of in-distribution forecasting performance.

**Result**   As shown in Figure 3, SATS consistently outperforms all competing methods. Compared to $\text{Moirai}_L$, the best baseline trained on clean data, $\text{SATS}_B$ achieves a **6.9%** improvement while using **only 22.6%** of its parameters. Similarly, against $\text{Chronos}_S$, the strongest baseline under data contamination, $\text{SATS}_S$ achieves superior performance with just **30.4%** of its parameter count. Notably, the gain from $\text{SATS}_S$ to $\text{SATS}_B$ is modest, likely because in-distribution forecasting involves limited temporal complexity, where increasing model size yields diminishing returns.

## 4.4 ABLATION STUDIES

**Module Design**   We begin by conducting ablation studies on the modules within $\text{SATS}_B$ to validate their effectiveness. As shown in Table 8, removing the Scale-aware Alignment leads to suboptimal performance, while discarding any component of the Hybrid Masking strategy results in further degradation. These results highlight the fundamental role of Hybrid

Table 3: Ablation study under the zero-shot evaluation setup. The averaged MSE and MAE are reported.

| Model variants | MSE | MAE |
|---|---|---|
| $\text{SATS}_B$ | **0.316** | **0.351** |
| w/o Scale-aware Alignment | 0.321 | 0.355 |
| w/o Continuous Masking | 0.338 | 0.362 |
| w/o Random Masking | 0.332 | 0.355 |

Masking in enhancing the training efficacy of encoder-only architectures, enabling the model to effectively capture diverse temporal scales. The Scale-aware Alignment offers additional performance improvements and complements this effect. Full results are provided in Appendix C.3.1.

**Alignment Mechanism**   The key design of Scale-aware Alignment is to minimize the distance between mean embeddings while maximizing the distance between maximal embeddings, thereby achieving alignment while preventing feature collapse. We further explore its mechanism by varying the pooling strategies involved, thereby offering empirical evidence for the selection of pooling methods. As shown in Table 9, removing the repulsion term between maximal embeddings leads to a significant performance drop, which is expected due to feature collapse. We then alter the pooling strategy used to define the embeddings whose distances are maximized: both min pooling and random pooling result in degraded performance, which indicates that maximal embeddings can more effectively encode scale-specific information in practice. Furthermore, applying alignment solely by minimizing the distance between maximal embeddings yields similarly suboptimal results to completely removing the alignment objective, suggesting that such a constraint is too weak to be effective. Full results are provided in Appendix C.3.2.

Table 4: Ablation study under the zero-shot evaluation setup. The averaged MSE and MAE are reported. "–" indicates that the corresponding training objective is removed.

| $\text{SATS}_B$ | | ETTh1 | | ETTh2 | | ETTm1 | | ETTm2 | | Weather | | Average | |
|---|---|---|---|---|---|---|---|---|---|---|---|---|---|
| Close | Far | MSE | MAE | MSE | MAE | MSE | MAE | MSE | MAE | MSE | MAE | MSE | MAE |
| Mean | Max | 0.395 | 0.414 | 0.334 | 0.377 | 0.362 | 0.373 | 0.250 | 0.309 | 0.239 | 0.283 | **0.316** | **0.351** |
| Mean | - | 0.409 | 0.423 | 0.367 | 0.397 | 0.393 | 0.390 | 0.312 | 0.345 | 0.269 | 0.294 | 0.350 | 0.370 |
| Mean | Min | 0.415 | 0.426 | 0.357 | 0.402 | 0.390 | 0.390 | 0.274 | 0.332 | 0.248 | 0.287 | 0.337 | 0.367 |
| Mean | Random | 0.400 | 0.416 | 0.344 | 0.390 | 0.356 | 0.375 | 0.268 | 0.328 | 0.243 | 0.288 | 0.322 | 0.359 |
| Max | - | 0.399 | 0.417 | 0.337 | 0.379 | 0.376 | 0.381 | 0.255 | 0.315 | 0.237 | 0.277 | 0.321 | 0.354 |
| - | - | 0.397 | 0.416 | 0.343 | 0.391 | 0.359 | 0.374 | 0.263 | 0.313 | 0.244 | 0.281 | 0.321 | 0.355 |

## 4.5 MODEL ANALYSIS

**T-SNE Visualization**   We visualize the token distributions of SATS and its without Scale-aware Alignment variant using t-SNE, as illustrated in Figure 4. Compared to the variant, SATS consistently exhibits superior token mapping, with a highly structured token space that yields clearly defined clusters in the t-SNE visualization. Notably, even in the second comparative setting where tokens from patch size 8 are extremely scarce, SATS still demonstrates robust scale-aware mapping. In contrast, although the without Scale-aware Alignment variant learns partially structured token

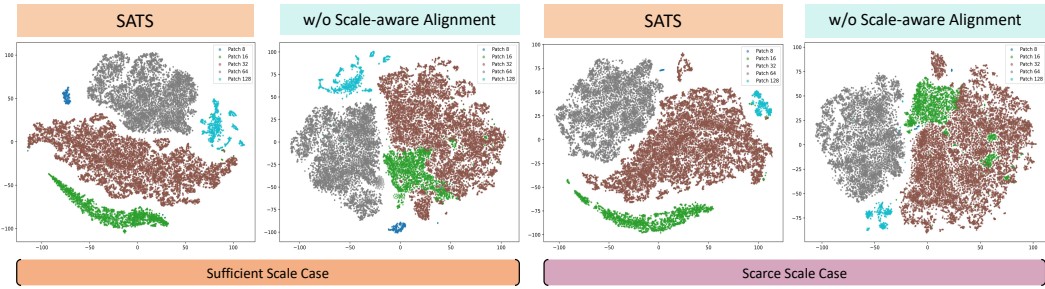

Figure 4: T-SNE visualization of token distributions under two regimes: **Sufficient Scale Case**, where each patch size retains a reasonable number of tokens, and **Scarce Scale Case**, where one or more patch sizes are extremely underrepresented. Colors indicate token origins from different patch sizes.

representations under large-scale training, it suffers from evident confusion between tokens from patch sizes 16 and 32, indicating a fragmented semantic space. Furthermore, when the number of tokens from patch size 8 is limited, these tokens are nearly overwhelmed, suggesting a complete loss of scale semantics during mapping. These empirical observations collectively underscore the effectiveness of the Scale-aware Alignment, which provides principled guidance for token generation. By ensuring semantic consistency across tokens, it enables the Transformer encoder to process more coherent representations, thereby enhancing the quality of model pretraining.

**Model Efficiency** Although the preceding discussions rarely highlight this aspect, both core techniques employed by SATS are parameter-free. This design choice enables SATS to achieve state-of-the-art performance with virtually no additional computational overhead. To more comprehensively reflect both predictive performance and resource usage, we introduce a model efficiency metric defined as the inverse of the product between the zero-shot error and the logarithm of model size. As illustrated in Figure 5, SATS demonstrates remarkable model efficiency. $SATS_B$ not only achieves SOTA accuracy but also surpasses the runner-up model, Timer-XL, by **8.9%** in efficiency. While $SATS_S$ achieves only slightly better performance than Timer-XL, it delivers a striking **70.1%** improvement in model effi-

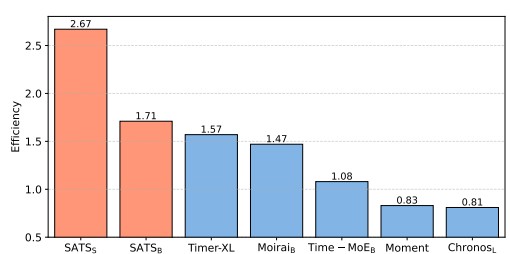

Figure 5: Model efficiency comparison based on a score defined as the inverse of MSE multiplied by the logarithm of parameter count. Higher values indicate better trade-offs between accuracy and model size. The MSE used here is the average reported in the zero-shot setting.

ciency. These results highlight the practical advantages of SATS—offering a compelling balance between accuracy and efficiency, making it particularly suitable for deployment in resource-constrained or real-time environments.

## 5 CONCLUSION

This paper presents SATS, a **S**cale-**A**ware foundation model for **T**ime **S**eries that addresses the challenge of fragmented token spaces and misaligned representations in time series pretraining. A scale-aware alignment mechanism is introduced to unify representations across patch sizes by jointly minimizing inter-scale embedding discrepancies and preserving scale-specific modeling capacity. Furthermore, a hybrid masking strategy combines random and contiguous masking to capture temporal dependencies at multiple resolutions. Extensive experiments demonstrate that SATS achieves superior generalization and robustness-—while remaining highly efficient due to its entirely parameter-free design.

## 6 ETHICS STATEMENT

Our work focuses on the pre-training of foundation models for time series forecasting, and therefore involves no potential ethical risks.

## 7 REPRODUCIBILITY STATEMENT

We provide a rigorous formulation of the model architecture in the main text, while deferring detailed implementation aspects—such as evaluation metrics, model specifications, and experimental setups—to the Appendix. To support reproducibility, we have submitted checkpoints of $SATS_S$ together with testing code for rapid validation. The full training code will be released publicly upon acceptance of the paper.

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

# A    THEORETICAL ANALYSIS OF MEAN VS. MAX STATISTICS

**Setup.**    Consider a time series decomposed as

$$x[n] = \ell[n] + h[n], \tag{10}$$

where (i) $\ell[n]$ is the low-frequency component satisfying a Lipschitz condition $|\ell[n] - \ell[m]| \leq K|n-m|$, and (ii) $h[n]$ is the high-frequency component with zero mean and variance $\sigma_h^2$. We focus on two statistics over a patch of length $L$:

$$\mu_L = \frac{1}{L} \sum_{n=1}^{L} x[n], \qquad M_L = \max_{1 \leq n \leq L} x[n]. \tag{11}$$

## A.1    MEAN STATISTIC: CROSS-PATCH CONSISTENCY

**Proposition A1 (Low-pass property).**    The mean operator $\mu_L$ is equivalent to convolution with a rectangular kernel, i.e.

$$\mu_L[n] = (x * w_L)[n], \quad w_L[k] = \frac{1}{L}\mathbf{1}_{\{0,\dots,L-1\}}(k), \tag{12}$$

with frequency response

$$|H_L(e^{j\omega})| = \left| \frac{\sin(\omega L/2)}{L \sin(\omega/2)} \right|. \tag{13}$$

Hence $\mu_L$ behaves as a low-pass filter, preserving the trend $\ell[n]$ while suppressing high-frequency variations $h[n]$.

**Proposition A2 (Variance decay).**    We can decompose

$$\mu_L = \frac{1}{L} \sum \ell[n] + \frac{1}{L} \sum h[n]. \tag{14}$$

Since $h[n]$ is zero-mean with variance $\sigma_h^2$, one obtains

$$\mathrm{Var}(\mu_L) \leq \frac{C\sigma_h^2}{L}. \tag{15}$$

Thus the variance of $\mu_L$ vanishes at rate $O(1/L)$, ensuring stability as patch length increases.

**Proposition A3 (Cross-scale expectation bound).**    For two patches with lengths $L_1, L_2$, the Lipschitz condition yields

$$\left| \mathbb{E}[\mu_{L_1}] - \mathbb{E}[\mu_{L_2}] \right| \leq \frac{K}{2}|L_1 - L_2|. \tag{16}$$

Therefore, the mean statistic exhibits bounded deviation across scales.

**Remark.**    Combining A2 and A3, the mean statistic $\mu_L$ is consistent across patches: expectation differences are small, variance decays with $L$, and the operator preserves low-frequency trends.

## A.2    MAX STATISTIC: CROSS-PATCH DISCRIMINABILITY

**Proposition B1 (High-frequency amplification).**    The max statistic can be written as

$$M_L = \max_{n \leq L}\{\ell[n] + h[n]\} \approx \ell[n^*] + \max_{n \leq L} h[n], \tag{17}$$

where $n^* = \arg\max x[n]$. The high-frequency component dominates the fluctuation of $M_L$. Classical extreme value theory implies

$$\mathbb{E}\left[ \max_{n \leq L} h[n] \right] \asymp \sigma_h \sqrt{2\log L}, \tag{18}$$

indicating that $M_L$ grows with $\sqrt{\log L}$ and is highly sensitive to high-frequency variation.

**Proposition B2 (Cross-scale separation).** For two patch lengths $L_1, L_2$, one can approximate

$$\mathbb{E}[M_{L_1}] - \mathbb{E}[M_{L_2}] \approx \ell(n_1^*) - \ell(n_2^*) + \sigma_h\left(\sqrt{2\log L_1} - \sqrt{2\log L_2}\right). \tag{19}$$

Hence cross-patch differences are amplified by the high-frequency component, scaling with $\sqrt{\log L}$.

**Remark.** The max statistic is discriminative: it accentuates local high-frequency peaks, leading to pronounced separation between patches of different lengths or positions.

### A.3 SUMMARY

Mean achieves cross-patch consistency by suppressing high-frequency variation ($O(1/L)$ variance decay), while max achieves discriminability by amplifying high-frequency differences (scale-dependent $\sqrt{\log L}$ growth).

## B EXPERIMENTAL DETAILS

### B.1 HARDWARE AND SOFTWARE CONFIGURATION

All variants of the SATS model were trained and evaluated on a single NVIDIA L40 GPU with 48 GB of VRAM. The system is powered by an Intel(R) Xeon(R) Platinum 8468V CPU and runs Ubuntu 20.04 LTS. The software stack includes Python 3.10 (managed via Miniconda) and PyTorch (Paszke et al., 2019) version 2.4.1.

Training was conducted using TensorFloat-32 (TF32) precision for applicable operations, in accordance with the default behavior of PyTorch on Ampere-generation GPUs.

### B.2 HYPERPARAMETER SETTINGS

All experiments use the following fixed hyperparameters unless otherwise specified:

- **Optimizer:** AdamW with learning rate $1 \times 10^{-3}$, weight decay $1 \times 10^{-1}$, $\beta_1 = 0.9$, $\beta_2 = 0.98$.
- **Scale-aware Alignment:** Temperature $\tau_{\text{mean}} = 0.1$ (Eq. 3), $\tau_{\text{max}} = 0.2$ (Eq. 4).
- **Hybrid Masking Strategy:** Masking probability $p = 0.5$ for balanced random and contiguous masking.
- **Loss Weights:** Primary objective weight $\alpha = 0.1$, auxiliary objective weight $\beta = 0.3$.

Due to limited computational resources and empirical evidence suggesting that large-scale language model pretraining is relatively robust to hyperparameter choices within reasonable ranges — as performance is primarily governed by scale rather than fine-tuned hyperparameters (Liu et al., 2019; Kaplan et al., 2020) — no further hyperparameter tuning was performed beyond the values listed above. Replacing empirical assumptions with rigorous empirical evidence is a necessary step for future work — we encourage systematic validation of these hyperparameter settings.

### B.3 EVALUATION METRICS

#### B.3.1 ZERO-SHOT FORECASTING

Following standard experimental protocols, we adopt Mean Squared Error (MSE) and Mean Absolute Error (MAE) as our primary evaluation metrics. These metrics are formulated as follows: [formulas to be inserted here].

$$\text{MSE} = \frac{1}{H} \sum_{h=1}^{H} \left(\mathbf{Y}_h - \widetilde{\mathbf{Y}}_h\right)^2, \tag{20}$$

$$\text{MAE} = \frac{1}{H} \sum_{h=1}^{H} \left|\mathbf{Y}_h - \widetilde{\mathbf{Y}}_h\right|, \tag{21}$$

Here, $Y_h$ and $\widetilde{Y}_h$ denote the $h$-th ground truth and predicted values, respectively, where $h \in 1, 2, ..., H$

### B.3.2 IN-DISTRIBUTION FORECASTING

We evaluate model performance on in-distribution forecasting using the Monash Time Series Forecasting Archive Godahewa et al. (2021). Due to the high variance in prediction scales across datasets, we follow the normalization protocol proposed by Woo et al., where the MAE is normalized using a naive forecast and then aggregated using the geometric mean. This procedure can be formalized as follows:

$$\text{N-MAE}_i = \frac{\text{MAE}_i}{\text{MAE}_i^{\text{naive}}} \tag{22}$$

$$\text{Result} = \left( \prod_{i=1}^{N} \text{N-MAE}_i \right)^{1/N} \tag{23}$$

where $\text{MAE}_i$ and $\text{MAE}_i^{\text{naive}}$ denote the MAE of the evaluated model and the naive baseline on the $i$-th dataset, respectively, and $N$ is the number of datasets.

### B.3.3 MODEL EFFICIENCY

Existing efficiency comparisons of pretrained models typically emphasize inference speed and runtime resource usage Wang et al. (2025); Liu et al. (2024b; 2025). While important, such evaluations often neglect training costs, which constitute a substantial portion of overall resource consumption. To provide a more comprehensive assessment, we propose an efficiency metric that integrates both resource usage (training + inference) and model generalization:

$$\text{Efficiency} = \frac{1}{\text{MSE}_{\text{zero-shot}} \times \log(\text{Params})} \tag{24}$$

Here, $\text{MSE}_{\text{zero-shot}}$ denotes the average mean squared error in zero-shot settings, and Params is the number of model parameters (in millions).

Using parameter count accounts for deployment cost, and applying a logarithmic scale moderates the effect of parameter size, emphasizing efficiency improvements that stem from architectural innovations rather than mere scale. We consider this a preliminary yet meaningful step toward more holistic evaluation of pretrained models.

### B.4 EVALUATION PROTOCOL AND ERROR BARS

Following Moirai, as described in the main text, we perform hyperparameter search over lookback window lengths {1000, 2000, 3000, 4000, 5000}, and over patch sizes determined by the dataset-specific mapping protocol proposed by Woo et al., which adapts patch sizes to the intrinsic sampling frequency of each dataset:

- Yearly, Quarterly: 8
- Monthly: 8, 16, 32
- Weekly, Daily: 16, 32
- Hourly: 32, 64
- Minute-level: 32, 64, 128
- Second-level: 64, 128

Although this protocol provides a range of hyperparameter options, we empirically choose the largest feasible patch sizes and lookback windows of at least 3000, as this tends to improve evaluation performance.

All reported results are based on 100 samples drawn from the predictive distribution, where we report the better of the mean and median for evaluation.

Some may suspect that searching input lengths only for SATS is unfair. However, pretrained models typically impose strict constraints on admissible input lengths. For instance, Time-MoE (Shi et al., 2024) requires the input length to be exactly four times the output length, while Timer-XL (Liu et al., 2025) selects the optimal input length depending on the dataset. Applying the same search protocol to these models would therefore be suboptimal. To ensure fairness, we adopt their original configurations and report their best results, thereby constructing a sufficiently competitive benchmark.

## C  DETAILED EXPERIMENTAL RESULTS

### C.1  ZERO-SHOT FORECASTING

We present the complete zero-shot forecasting results to complement the main text. Specifically, we construct the zero-shot benchmark based on five widely used datasets: ETTh1, ETTh2, ETTm1, ETTm2, and Weather. Two other datasets, ECL and Traffic, which are popular choices in small-scale model evaluations, are excluded here since **they are already included in most pre-training corpora, and their usage would thus compromise the fairness of a comprehensive leaderboard**. Overall, adopting these five datasets strikes a balance and serves as the greatest common ground for zero-shot evaluation. As shown in Table 5, all SATS variants consistently outperform their competitors, demonstrating superior generalization ability and robust performance across diverse datasets. In addition, SATS exhibits a clear performance gain as model size increases, revealing strong scalability. This trend contrasts with models such as Time-MoE (Shi et al., 2024) and Moirai (Woo et al., 2024), whose performance plateaus or even degrades with larger model configurations.

Table 5: Full results of zero-shot forecasting across all evaluated models. Lower values of MSE and MAE indicate superior performance. As TimesFM incorporates Weather data during pretraining, it is excluded from evaluation on this dataset (denoted by "–").

| Models | | $SATS_S$ | | $SATS_B$ | | Timer-XL | | $Time\text{-}MoE_L$ | | $Time\text{-}MoE_B$ | | $Moirai_T$ | | $Moirai_B$ | | $Moirai_S$ | | $Chronos_L$ | | $Chronos_B$ | | $Chronos_S$ | | Moment | | TimesFM | |
|---|---|---|---|---|---|---|---|---|---|---|---|---|---|---|---|---|---|---|---|---|---|---|---|---|---|---|---|---|
| Metrics | | MSE | MAE | MSE | MAE | MSE | MAE | MSE | MAE | MSE | MAE | MSE | MAE | MSE | MAE | MSE | MAE | MSE | MAE | MSE | MAE | MSE | MAE | MSE | MAE | MSE | MAE |
| ETTh1 | 96 | 0.375 | 0.393 | 0.360 | 0.387 | 0.369 | 0.391 | 0.350 | 0.382 | 0.357 | 0.381 | 0.381 | 0.398 | 0.383 | 0.402 | 0.375 | 0.402 | 0.441 | 0.390 | 0.440 | 0.393 | 0.466 | 0.409 | 0.688 | 0.557 | 0.414 | 0.404 |
| | 192 | 0.412 | 0.415 | 0.395 | 0.409 | 0.405 | 0.413 | 0.388 | 0.412 | 0.384 | 0.404 | 0.428 | 0.427 | 0.425 | 0.429 | 0.399 | 0.419 | 0.502 | 0.424 | 0.492 | 0.426 | 0.530 | 0.450 | 0.688 | 0.560 | 0.465 | 0.434 |
| | 336 | 0.423 | 0.425 | 0.413 | 0.422 | 0.418 | 0.423 | 0.411 | 0.430 | 0.411 | 0.434 | 0.458 | 0.456 | 0.450 | 0.450 | 0.412 | 0.429 | 0.576 | 0.467 | 0.550 | 0.462 | 0.570 | 0.486 | 0.675 | 0.563 | 0.503 | 0.456 |
| | 720 | 0.418 | 0.441 | 0.413 | 0.438 | 0.423 | 0.441 | 0.427 | 0.455 | 0.449 | 0.477 | 0.502 | 0.477 | 0.470 | 0.473 | 0.413 | 0.444 | 0.835 | 0.583 | 0.882 | 0.591 | 0.615 | 0.543 | 0.683 | 0.585 | 0.511 | 0.481 |
| | AVG | 0.407 | 0.418 | 0.395 | 0.414 | 0.404 | 0.417 | 0.394 | 0.420 | 0.400 | 0.424 | 0.442 | 0.437 | 0.433 | 0.438 | 0.400 | 0.424 | 0.589 | 0.466 | 0.591 | 0.468 | 0.545 | 0.472 | 0.684 | 0.566 | 0.473 | 0.444 |
| ETTh2 | 96 | 0.283 | 0.328 | 0.273 | 0.331 | 0.283 | 0.342 | 0.302 | 0.354 | 0.305 | 0.359 | 0.287 | 0.329 | 0.277 | 0.327 | 0.281 | 0.334 | 0.320 | 0.345 | 0.308 | 0.343 | 0.307 | 0.356 | 0.342 | 0.396 | 0.315 | 0.349 |
| | 192 | 0.343 | 0.369 | 0.330 | 0.372 | 0.340 | 0.379 | 0.364 | 0.385 | 0.351 | 0.386 | 0.349 | 0.372 | 0.340 | 0.374 | 0.340 | 0.373 | 0.406 | 0.399 | 0.384 | 0.392 | 0.376 | 0.401 | 0.354 | 0.402 | 0.388 | 0.395 |
| | 336 | 0.365 | 0.391 | 0.353 | 0.396 | 0.366 | 0.400 | 0.417 | 0.425 | 0.391 | 0.418 | 0.372 | 0.392 | 0.371 | 0.401 | 0.362 | 0.393 | 0.492 | 0.453 | 0.429 | 0.430 | 0.408 | 0.431 | 0.356 | 0.407 | 0.422 | 0.427 |
| | 720 | 0.404 | 0.424 | 0.380 | 0.409 | 0.397 | 0.431 | 0.537 | 0.496 | 0.419 | 0.454 | 0.403 | 0.423 | 0.394 | 0.426 | 0.380 | 0.416 | 0.603 | 0.511 | 0.501 | 0.477 | 0.604 | 0.533 | 0.395 | 0.434 | 0.443 | 0.454 |
| | AVG | 0.349 | 0.378 | 0.334 | 0.377 | 0.347 | 0.388 | 0.405 | 0.415 | 0.367 | 0.404 | 0.353 | 0.379 | 0.345 | 0.382 | 0.341 | 0.379 | 0.455 | 0.427 | 0.406 | 0.411 | 0.424 | 0.430 | 0.362 | 0.410 | 0.392 | 0.406 |
| ETTm1 | 96 | 0.325 | 0.353 | 0.323 | 0.345 | 0.317 | 0.356 | 0.309 | 0.357 | 0.338 | 0.368 | 0.612 | 0.444 | 0.396 | 0.382 | 0.495 | 0.409 | 0.457 | 0.403 | 0.454 | 0.408 | 0.511 | 0.423 | 0.654 | 0.527 | 0.361 | 0.370 |
| | 192 | 0.352 | 0.372 | 0.352 | 0.364 | 0.358 | 0.381 | 0.346 | 0.381 | 0.353 | 0.388 | 0.593 | 0.446 | 0.425 | 0.402 | 0.548 | 0.431 | 0.530 | 0.450 | 0.567 | 0.477 | 0.618 | 0.485 | 0.662 | 0.532 | 0.414 | 0.405 |
| | 336 | 0.372 | 0.387 | 0.371 | 0.379 | 0.386 | 0.401 | 0.373 | 0.408 | 0.381 | 0.413 | 0.591 | 0.454 | 0.452 | 0.415 | 0.577 | 0.445 | 0.577 | 0.481 | 0.662 | 0.525 | 0.683 | 0.524 | 0.672 | 0.537 | 0.445 | 0.429 |
| | 720 | 0.405 | 0.410 | 0.401 | 0.403 | 0.430 | 0.431 | 0.475 | 0.477 | 0.504 | 0.493 | 0.596 | 0.468 | 0.477 | 0.431 | 0.586 | 0.457 | 0.660 | 0.526 | 0.900 | 0.591 | 0.748 | 0.566 | 0.692 | 0.551 | 0.512 | 0.471 |
| | AVG | 0.364 | 0.380 | 0.362 | 0.373 | 0.373 | 0.392 | 0.376 | 0.406 | 0.394 | 0.416 | 0.598 | 0.453 | 0.437 | 0.407 | 0.551 | 0.436 | 0.556 | 0.465 | 0.646 | 0.500 | 0.640 | 0.500 | 0.670 | 0.537 | 0.433 | 0.419 |
| ETTm2 | 96 | 0.172 | 0.255 | 0.167 | 0.251 | 0.189 | 0.277 | 0.197 | 0.286 | 0.201 | 0.291 | 0.189 | 0.260 | 0.195 | 0.269 | 0.211 | 0.290 | 0.197 | 0.271 | 0.199 | 0.274 | 0.209 | 0.291 | 0.260 | 0.335 | 0.202 | 0.270 |
| | 192 | 0.226 | 0.292 | 0.222 | 0.290 | 0.241 | 0.315 | 0.250 | 0.322 | 0.258 | 0.334 | 0.247 | 0.300 | 0.247 | 0.303 | 0.264 | 0.325 | 0.254 | 0.314 | 0.261 | 0.322 | 0.280 | 0.341 | 0.289 | 0.350 | 0.289 | 0.321 |
| | 336 | 0.279 | 0.327 | 0.269 | 0.323 | 0.286 | 0.348 | 0.337 | 0.375 | 0.324 | 0.373 | 0.295 | 0.334 | 0.291 | 0.333 | 0.312 | 0.356 | 0.313 | 0.353 | 0.326 | 0.366 | 0.354 | 0.390 | 0.324 | 0.369 | 0.360 | 0.366 |
| | 720 | 0.369 | 0.385 | 0.343 | 0.374 | 0.375 | 0.402 | 0.480 | 0.461 | 0.488 | 0.464 | 0.372 | 0.386 | 0.355 | 0.377 | 0.395 | 0.405 | 0.416 | 0.415 | 0.455 | 0.439 | 0.553 | 0.499 | 0.394 | 0.409 | 0.462 | 0.430 |
| | AVG | 0.262 | 0.315 | 0.250 | 0.309 | 0.273 | 0.336 | 0.316 | 0.361 | 0.318 | 0.366 | 0.276 | 0.320 | 0.272 | 0.321 | 0.295 | 0.344 | 0.295 | 0.338 | 0.310 | 0.350 | 0.349 | 0.380 | 0.317 | 0.366 | 0.328 | 0.347 |
| Weather | 96 | 0.180 | 0.236 | 0.162 | 0.217 | 0.189 | 0.225 | 0.159 | 0.213 | 0.160 | 0.214 | 0.174 | 0.204 | 0.175 | 0.210 | 0.173 | 0.212 | 0.194 | 0.235 | 0.203 | 0.238 | 0.211 | 0.243 | 0.243 | 0.255 | - | - |
| | 192 | 0.226 | 0.280 | 0.210 | 0.265 | 0.221 | 0.271 | 0.215 | 0.266 | 0.210 | 0.260 | 0.221 | 0.248 | 0.218 | 0.251 | 0.216 | 0.250 | 0.249 | 0.285 | 0.256 | 0.290 | 0.263 | 0.294 | 0.278 | 0.329 | - | - |
| | 336 | 0.274 | 0.316 | 0.258 | 0.302 | 0.274 | 0.311 | 0.291 | 0.322 | 0.274 | 0.309 | 0.271 | 0.287 | 0.267 | 0.288 | 0.260 | 0.282 | 0.302 | 0.327 | 0.314 | 0.336 | 0.321 | 0.339 | 0.306 | 0.346 | - | - |
| | 720 | 0.341 | 0.363 | 0.325 | 0.349 | 0.356 | 0.370 | 0.415 | 0.400 | 0.418 | 0.405 | 0.340 | 0.332 | 0.338 | 0.338 | 0.320 | 0.322 | 0.372 | 0.378 | 0.397 | 0.396 | 0.404 | 0.397 | 0.350 | 0.374 | - | - |
| | AVG | 0.255 | 0.299 | 0.239 | 0.283 | 0.256 | 0.294 | 0.270 | 0.300 | 0.266 | 0.297 | 0.251 | 0.268 | 0.250 | 0.271 | 0.242 | 0.267 | 0.279 | 0.306 | 0.293 | 0.315 | 0.300 | 0.318 | 0.294 | 0.326 | - | - |
| Average | | 0.327 | 0.358 | 0.316 | 0.351 | 0.330 | 0.365 | 0.352 | 0.380 | 0.349 | 0.381 | 0.384 | 0.371 | 0.348 | 0.364 | 0.366 | 0.370 | 0.435 | 0.401 | 0.449 | 0.409 | 0.452 | 0.420 | 0.465 | 0.441 | - | - |

### C.2  IN-DISTRIBUTION FORECASTING

We present the detailed results of all foundation models and baselines on the Monash benchmark Godahewa et al. (2021). As shown in Tables 6 and 7, we report the performance on each individual dataset, including the normalized MAE and the aggregated GEOMEAN adopted in the main text. Given the wide heterogeneity across the Monash datasets, we focus on comparing the final aggregated metrics. The results demonstrate that all variants of SATS consistently outperform the competing methods.

Table 6: Full in-distribution forecasting results of foundation models on the Monash benchmark Godahewa et al. (2021). NMAE-N denotes the MAE normalized by the naive forecast, and GEOMEAN represents the geometric mean across all series.

| Model | SATS$_S$ | | SATS$_B$ | | Moirai$_S$ | | Moirai$_B$ | | Moirai$_L$ | | Chronos$_S$ | | Chronos$_B$ | | Chronos$_L$ | | LLMTime | | TimesFM | | Naive |
|---|---|---|---|---|---|---|---|---|---|---|---|---|---|---|---|---|---|---|---|---|---|
| Metrics | MAE | NMAE-N | MAE | NMAE-N | MAE | NMAE-N | MAE | NMAE-N | MAE | NMAE-N | MAE | NMAE-N | MAE | NMAE-N | MAE | NMAE-N | MAE | NMAE-N | MAE | NMAE-N | MAE |
| M1 Monthly | 1950.16 | 0.72 | 2072.65 | 0.77 | 2082.26 | 0.77 | 2068.63 | 0.76 | 1983.18 | 0.73 | 1797.78 | 0.66 | 1637.68 | 0.60 | 1627.11 | 0.60 | 2562.84 | 0.95 | 1673.60 | 0.62 | 2707.75 |
| M3 Monthly | 686.47 | 0.82 | 668.78 | 0.80 | 713.41 | 0.85 | 658.17 | 0.79 | 664.03 | 0.79 | 644.38 | 0.77 | 622.27 | 0.74 | 619.79 | 0.74 | 877.97 | 1.05 | 653.57 | 0.78 | 837.14 |
| M3 Other | 230.95 | 0.83 | 205.19 | 0.74 | 263.54 | 0.95 | 198.62 | 0.71 | 202.41 | 0.73 | 196.59 | 0.71 | 191.80 | 0.69 | 205.93 | 0.74 | 300.30 | 1.08 | 207.23 | 0.74 | 278.43 |
| M4 Monthly | 596.94 | 0.89 | 587.60 | 0.88 | 597.60 | 0.89 | 592.09 | 0.88 | 584.36 | 0.87 | 592.85 | 0.88 | 598.46 | 0.89 | 584.78 | 0.87 | 728.27 | 1.08 | 580.20 | 0.86 | 671.27 |
| M4 Weekly | 323.21 | 0.93 | 322.21 | 0.93 | 339.76 | 0.98 | 328.08 | 0.94 | 301.52 | 0.87 | 264.56 | 0.76 | 252.26 | 0.72 | 248.89 | 0.72 | 518.44 | 1.49 | 285.89 | 0.82 | 347.99 |
| M4 Daily | 173.44 | 0.96 | 185.84 | 1.03 | 189.10 | 1.05 | 192.66 | 1.07 | 189.78 | 1.05 | 169.91 | 0.94 | 177.49 | 0.98 | 168.41 | 0.93 | 266.52 | 1.47 | 172.98 | 0.96 | 180.83 |
| M4 Hourly | 190.61 | 0.16 | 242.96 | 0.20 | 268.04 | 0.22 | 209.87 | 0.17 | 197.79 | 0.16 | 214.18 | 0.18 | 230.70 | 0.19 | 201.14 | 0.17 | 576.06 | 0.47 | 196.20 | 0.16 | 1218.00 |
| Tourism Quarterly | 7853.84 | 0.50 | 8618.77 | 0.54 | 18352.44 | 1.16 | 17196.86 | 1.09 | 15820.02 | 1.00 | 7823.27 | 0.49 | 8835.52 | 0.56 | 8521.70 | 0.54 | 16918.86 | 1.07 | 10568.92 | 0.67 | 15845.10 |
| Tourism Monthly | 2710.72 | 0.48 | 2579.48 | 0.46 | 3569.85 | 0.63 | 2862.06 | 0.51 | 2688.55 | 0.48 | 2465.10 | 0.44 | 2358.67 | 0.42 | 2140.73 | 0.38 | 5608.61 | 0.99 | 2422.01 | 0.43 | 5636.83 |
| CIF 2016 | 504502.50 | 0.87 | 521981.25 | 0.90 | 655888.58 | 1.13 | 539222.03 | 0.93 | 695156.92 | 1.20 | 649110.99 | 1.12 | 604088.54 | 1.04 | 728981.15 | 1.26 | 599313.84 | 1.04 | 819922.44 | 1.42 | 578596.53 |
| Bitcoin | 8.20E+17 | 1.05 | 7.61E+17 | 0.98 | 1.76E+18 | 2.26 | 1.62E+18 | 2.08 | 1.87E+18 | 2.40 | 2.34E+18 | 3.01 | 2.27E+18 | 2.92 | 1.88E+18 | 2.42 | 1.74E+18 | 2.236503856 | 7.78E+17 | 1.00 | 7.78E+17 |
| Pedestrian Counts | 48.94 | 0.29 | 47.85 | 0.28 | 54.88 | 0.32 | 54.08 | 0.32 | 41.66 | 0.24 | 29.77 | 0.17 | 27.34 | 0.16 | 26.95 | 0.16 | 97.77 | 0.57 | 45.03 | 0.26 | 170.88 |
| Vehicle Trips | 20.20 | 0.64 | 20.79 | 0.66 | 24.46 | 0.78 | 23.17 | 0.74 | 21.85 | 0.70 | 19.38 | 0.62 | 19.25 | 0.61 | 19.19 | 0.61 | 31.48 | 1.00 | 21.93 | 0.70 | 31.42 |
| KDD cup | 38.69 | 0.92 | 37.00 | 0.88 | 39.81 | 0.94 | 38.66 | 0.92 | 39.09 | 0.93 | 38.60 | 0.92 | 42.36 | 1.01 | 38.83 | 0.92 | 42.72 | 1.01 | 40.86 | 0.97 | 42.13 |
| Weather | 1.89 | 0.80 | 1.89 | 0.80 | 1.96 | 0.83 | 1.80 | 0.76 | 1.75 | 0.74 | 1.96 | 0.83 | 1.84 | 0.78 | 1.85 | 0.78 | 2.17 | 0.92 | 2.07 | 0.88 | 2.36 |
| NN5 Daily | 4.06 | 0.49 | 3.91 | 0.47 | 5.37 | 0.65 | 4.26 | 0.52 | 3.77 | 0.46 | 3.83 | 0.46 | 3.67 | 0.44 | 3.53 | 0.43 | 7.10 | 0.86 | 3.85 | 0.47 | 8.26 |
| NN5 Weekly | 14.63 | 0.88 | 14.72 | 0.88 | 15.07 | 0.90 | 16.42 | 0.98 | 15.30 | 0.92 | 15.03 | 0.90 | 15.12 | 0.90 | 15.09 | 0.90 | 15.76 | 0.94 | 15.09 | 0.90 | 16.71 |
| Carparts | 0.45 | 0.69 | 0.45 | 0.70 | 0.53 | 0.82 | 0.47 | 0.72 | 0.49 | 0.75 | 0.52 | 0.80 | 0.54 | 0.83 | 0.53 | 0.82 | 0.44 | 0.68 | 0.50 | 0.77 | 0.65 |
| FRED-MD | 2474.34 | 0.88 | 1511.46 | 0.53 | 2568.48 | 0.91 | 2679.29 | 0.95 | 2792.55 | 0.99 | 938.46 | 0.33 | 1036.67 | 0.37 | 863.99 | 0.31 | 2804.64 | 0.99 | 2237.63 | 0.79 | 2825.67 |
| Traffic Hourly | 0.02 | 0.51 | 0.01 | 0.50 | 0.02 | 0.67 | 0.02 | 0.67 | 0.01 | 0.33 | 0.01 | 0.43 | 0.01 | 0.40 | 0.01 | 0.33 | 0.03 | 1.00 | 0.01 | 0.30 | 0.03 |
| Traffic Weekly | 1.13 | 0.95 | 1.13 | 0.95 | 1.17 | 0.98 | 1.14 | 0.96 | 1.13 | 0.95 | 1.14 | 0.96 | 1.12 | 0.94 | 1.12 | 0.94 | 1.15 | 0.97 | 1.06 | 0.89 | 1.19 |
| Rideshare | 1.48 | 0.23 | 1.14 | 0.18 | 1.35 | 0.21 | 1.39 | 0.22 | 1.29 | 0.21 | 1.27 | 0.20 | 1.33 | 0.21 | 1.30 | 0.21 | 6.28 | 1.00 | 1.36 | 0.22 | 6.29 |
| Hospital | 19.64 | 0.82 | 18.57 | 0.77 | 23.00 | 0.96 | 19.40 | 0.81 | 19.44 | 0.81 | 19.74 | 0.82 | 19.75 | 0.82 | 19.88 | 0.83 | 25.68 | 1.07 | 18.54 | 0.77 | 24.07 |
| COVID Deaths | 98.28 | 0.28 | 118.60 | 0.34 | 124.32 | 0.35 | 126.11 | 0.36 | 117.11 | 0.33 | 207.47 | 0.59 | 118.26 | 0.33 | 190.01 | 0.54 | 653.31 | 1.85 | 623.47 | 1.76 | 353.71 |
| Temperature Rain | 5.21 | 0.55 | 5.24 | 0.56 | 5.30 | 0.56 | 5.08 | 0.54 | 5.27 | 0.56 | 5.35 | 0.57 | 5.17 | 0.55 | 5.19 | 0.55 | 6.37 | 0.68 | 5.27 | 0.56 | 9.39 |
| Sunspot | 0.09 | 0.02 | 0.12 | 0.03 | 0.11 | 0.03 | 0.08 | 0.02 | 0.13 | 0.03 | 0.20 | 0.05 | 2.45 | 0.62 | 3.45 | 0.88 | 5.07 | 1.29 | 1.07 | 0.27 | 3.93 |
| Saugeen River Flow | 22.52 | 1.05 | 20.59 | 0.96 | 24.07 | 1.12 | 24.40 | 1.13 | 24.76 | 1.15 | 23.57 | 1.10 | 25.54 | 1.19 | 26.25 | 1.22 | 34.84 | 1.62 | 25.16 | 1.17 | 21.50 |
| US Births | 466.23 | 0.40 | 507.99 | 0.44 | 872.51 | 0.76 | 624.30 | 0.54 | 476.50 | 0.41 | 432.14 | 0.37 | 420.08 | 0.36 | 432.14 | 0.37 | 1374.99 | 1.19 | 461.58 | 0.40 | 1152.67 |
| GEOMEAN | 199.95 | 0.55 | 197.66 | 0.54 | 239.80 | 0.66 | 218.28 | 0.60 | 210.24 | 0.58 | 204.67 | 0.56 | 217.23 | 0.60 | 220.03 | 0.60 | 380.04 | 1.04 | 235.10 | 0.64 | 365.08 |

Table 7: Full in-distribution forecasting results of baselines on the Monash benchmark Godahewa et al. (2021). NMAE-N denotes the MAE normalized by the naive forecast, and GEOMEAN represents the geometric mean across all series.

| Model | SES | | Theta | | TBATS | | ETS | | (DHR-)ARIMA | | PR | | CatBoost | | FFNN | | DeepAR | | N-BEATS | | WaveNet | | Transformer | | Naive |
|---|---|---|---|---|---|---|---|---|---|---|---|---|---|---|---|---|---|---|---|---|---|---|---|---|---|
| Metrics | MAE | NMAE-N | MAE | NMAE-N | MAE | NMAE-N | MAE | NMAE-N | MAE | NMAE-N | MAE | NMAE-N | MAE | NMAE-N | MAE | NMAE-N | MAE | NMAE-N | MAE | NMAE-N | MAE | NMAE-N | MAE | NMAE-N | MAE |
| M1 Monthly | 2259.04 | 0.83 | 2166.18 | 0.80 | 2237.50 | 0.83 | 1905.28 | 0.70 | 2080.13 | 0.77 | 2088.25 | 0.77 | 2052.32 | 0.76 | 2162.58 | 0.80 | 1860.81 | 0.69 | 1820.37 | 0.67 | 2184.42 | 0.81 | 2223.88 | 1.01 | 2707.75 |
| M3 Monthly | 743.41 | 0.89 | 623.71 | 0.75 | 630.59 | 0.75 | 626.46 | 0.75 | 654.80 | 0.78 | 692.97 | 0.83 | 732.00 | 0.87 | 692.48 | 0.83 | 728.81 | 0.87 | 648.60 | 0.77 | 699.30 | 0.84 | 798.38 | 0.95 | 837.14 |
| M3 Other | 277.83 | 1.00 | 215.35 | 0.77 | 189.42 | 0.68 | 194.98 | 0.70 | 193.02 | 0.69 | 234.43 | 0.84 | 318.13 | 1.14 | 240.17 | 0.86 | 247.56 | 0.89 | 221.85 | 0.80 | 245.29 | 0.88 | 239.24 | 0.86 | 278.43 |
| M4 Monthly | 625.24 | 0.93 | 563.58 | 0.84 | 589.52 | 0.88 | 582.60 | 0.87 | 575.36 | 0.86 | 596.19 | 0.89 | 611.69 | 0.91 | 612.52 | 0.91 | 615.22 | 0.92 | 578.48 | 0.86 | 655.51 | 0.98 | 780.47 | 1.16 | 671.27 |
| M4 Weekly | 336.82 | 0.97 | 333.32 | 0.96 | 296.15 | 0.85 | 335.66 | 0.96 | 321.61 | 0.92 | 293.21 | 0.84 | 364.65 | 1.05 | 338.37 | 0.97 | 351.78 | 1.01 | 277.73 | 0.80 | 399.46 | 1.03 | 378.89 | 1.09 | 347.99 |
| M4 Daily | 178.27 | 0.99 | 178.86 | 0.99 | 176.60 | 0.98 | 193.26 | 1.07 | 179.67 | 0.99 | 181.92 | 1.01 | 231.36 | 1.28 | 177.91 | 0.98 | 299.79 | 1.66 | 190.44 | 1.05 | 189.47 | 1.05 | 201.08 | 1.11 | 180.83 |
| M4 Hourly | 1218.06 | 1.00 | 1220.97 | 1.00 | 386.27 | 0.32 | 3358.10 | 2.76 | 1310.85 | 1.08 | 257.39 | 0.21 | 285.35 | 0.23 | 385.49 | 0.32 | 886.02 | 0.73 | 425.75 | 0.35 | 393.63 | 0.32 | 320.54 | 0.26 | 1218.06 |
| Tourism Quarterly | 15014.19 | 0.95 | 7656.49 | 0.48 | 9972.42 | 0.63 | 8925.52 | 0.56 | 10475.47 | 0.66 | 9092.58 | 0.57 | 10267.97 | 0.65 | 8981.04 | 0.57 | 9511.37 | 0.60 | 8640.56 | 0.55 | 9137.12 | 0.58 | 9521.67 | 0.60 | 15845.10 |
| Tourism Monthly | 5302.10 | 0.94 | 2069.96 | 0.37 | 2940.08 | 0.52 | 2004.51 | 0.36 | 2536.77 | 0.45 | 2187.28 | 0.39 | 2537.04 | 0.45 | 2022.21 | 0.36 | 1871.69 | 0.33 | 2003.02 | 0.36 | 2095.13 | 0.37 | 2146.98 | 0.38 | 5636.83 |
| CIF 2016 | 581875.97 | 1.01 | 714818.58 | 1.24 | 855578.40 | 1.48 | 643421.42 | 1.11 | 469059.49 | 0.81 | 563205.57 | 0.97 | 603551.30 | 1.04 | 1495923.44 | 2.59 | 320018.00 | 5.53 | 679034.80 | 1.17 | 599822.62 | 10.37 | 4057973.00 | 7.01 | 578596.53 |
| Aus. Elec. Demand | 659.60 | 1.00 | 665.04 | 1.01 | 370.74 | 0.56 | 1282.99 | 1.95 | 1045.92 | 1.59 | 247.18 | 0.37 | 241.77 | 0.37 | 258.76 | 0.39 | 302.41 | 0.46 | 213.83 | 0.32 | 227.50 | 0.34 | 231.45 | 0.35 | 659.60 |
| Bitcoin | 5.33E+18 | 6.85 | 5.33E+18 | 6.85 | 9.90E+17 | 1.27 | 1.10E+18 | 1.41 | 3.62E+18 | 4.65 | 6.66E+17 | 0.86 | 1.93E+18 | 2.48 | 1.45E+18 | 1.86 | 1.95E+18 | 2.51 | 1.06E+18 | 1.36 | 2.46E+18 | 3.16 | 2.61E+18 | 3.35 | 7.78E+17 |
| Pedestrian Counts | 170.87 | 1.00 | 170.94 | 1.00 | 222.38 | 1.30 | 216.50 | 1.27 | 635.16 | 3.72 | 44.18 | 0.26 | 43.41 | 0.25 | 46.41 | 0.27 | 44.78 | 0.26 | 66.84 | 0.39 | 46.46 | 0.27 | 47.29 | 0.28 | 170.88 |
| Vehicle Trips | 29.98 | 0.95 | 30.76 | 0.98 | 21.21 | 0.68 | 30.95 | 0.99 | 30.07 | 0.96 | 27.24 | 0.87 | 22.61 | 0.72 | 22.93 | 0.73 | 22.00 | 0.70 | 28.16 | 0.90 | 24.15 | 0.77 | 28.01 | 0.89 | 31.42 |
| KDD cup | 42.04 | 1.00 | 42.06 | 1.00 | 39.20 | 0.93 | 44.88 | 1.07 | 52.20 | 1.24 | 36.85 | 0.87 | 34.82 | 0.83 | 37.16 | 0.88 | 48.98 | 1.16 | 49.10 | 1.17 | 37.08 | 0.88 | 44.46 | 1.06 | 42.13 |
| Weather | 2.24 | 0.95 | 2.51 | 1.06 | 2.30 | 0.97 | 2.35 | 1.00 | 2.45 | 1.04 | 8.17 | 3.46 | 2.51 | 1.06 | 2.09 | 0.89 | 2.02 | 0.86 | 2.34 | 0.99 | 2.29 | 0.97 | 2.03 | 0.86 | 2.36 |
| NN5 Daily | 6.63 | 0.80 | 3.80 | 0.46 | 3.70 | 0.45 | 3.72 | 0.45 | 4.41 | 0.53 | 5.47 | 0.66 | 4.22 | 0.51 | 4.06 | 0.49 | 3.94 | 0.48 | 4.92 | 0.60 | 3.97 | 0.48 | 4.16 | 0.50 | 8.26 |
| NN5 Weekly | 15.66 | 0.94 | 15.30 | 0.92 | 14.98 | 0.90 | 15.70 | 0.94 | 15.38 | 0.92 | 14.94 | 0.89 | 15.29 | 0.92 | 15.02 | 0.90 | 14.69 | 0.88 | 14.19 | 0.85 | 19.34 | 1.16 | 20.34 | 1.22 | 16.71 |
| Carparts | 0.55 | 0.85 | 0.53 | 0.82 | 0.58 | 0.89 | 0.56 | 0.86 | 0.56 | 0.86 | 0.41 | 0.63 | 0.53 | 0.82 | 0.39 | 0.60 | 0.39 | 0.60 | 0.98 | 1.51 | 0.40 | 0.62 | 0.39 | 0.60 | 0.65 |
| FRED-MD | 2798.22 | 0.99 | 3492.84 | 1.24 | 1989.97 | 0.70 | 2041.42 | 0.72 | 2957.11 | 1.05 | 8921.94 | 3.16 | 2475.68 | 0.88 | 2239.57 | 0.83 | 4264.36 | 1.51 | 2557.80 | 0.91 | 2508.40 | 0.89 | 4666.04 | 1.65 | 2825.67 |
| Traffic Hourly | 0.03 | 1.00 | 0.03 | 1.00 | 0.04 | 1.33 | 0.03 | 1.00 | 0.04 | 1.33 | 0.02 | 0.67 | 0.02 | 0.67 | 0.01 | 0.33 | 0.01 | 0.33 | 0.02 | 0.67 | 0.02 | 0.67 | 0.01 | 0.33 | 0.03 |
| Traffic Weekly | 1.12 | 0.94 | 1.13 | 0.95 | 1.17 | 0.98 | 1.14 | 0.96 | 1.22 | 1.03 | 1.13 | 0.95 | 1.17 | 0.98 | 1.15 | 0.97 | 1.18 | 0.99 | 1.11 | 0.93 | 1.20 | 1.01 | 1.42 | 1.19 | 1.19 |
| Rideshare | 6.29 | 1.00 | 7.62 | 1.21 | 6.45 | 1.03 | 6.29 | 1.00 | 3.37 | 0.54 | 6.30 | 1.00 | 6.07 | 0.97 | 6.59 | 1.05 | 6.28 | 1.00 | 5.55 | 0.88 | 2.75 | 0.44 | 6.29 | 1.00 | 6.29 |
| Hospital | 21.76 | 0.90 | 18.54 | 0.77 | 17.43 | 0.72 | 17.97 | 0.75 | 19.60 | 0.81 | 19.24 | 0.80 | 19.17 | 0.80 | 22.86 | 0.95 | 18.25 | 0.76 | 20.18 | 0.84 | 19.35 | 0.80 | 36.19 | 1.50 | 24.07 |
| COVID Deaths | 353.71 | 1.00 | 321.32 | 0.91 | 96.29 | 0.27 | 85.59 | 0.24 | 85.77 | 0.24 | 347.98 | 0.98 | 475.15 | 1.34 | 144.14 | 0.41 | 201.98 | 0.57 | 158.81 | 0.45 | 1049.48 | 2.97 | 408.66 | 1.16 | 353.71 |
| Temperature Rain | 8.18 | 0.87 | 8.22 | 0.88 | 7.14 | 0.76 | 8.21 | 0.87 | 7.19 | 0.77 | 6.13 | 0.65 | 6.76 | 0.72 | 5.56 | 0.59 | 5.37 | 0.57 | 7.28 | 0.78 | 5.81 | 0.62 | 5.24 | 0.56 | 9.39 |
| Sunspot | 4.93 | 1.25 | 4.93 | 1.25 | 2.57 | 0.65 | 4.93 | 1.25 | 2.57 | 0.65 | 3.83 | 0.97 | 2.27 | 0.58 | 7.97 | 2.03 | 0.77 | 0.20 | 14.47 | 3.68 | 0.17 | 0.04 | 0.13 | 0.03 | 3.93 |
| Saugeen River Flow | 21.50 | 1.00 | 21.49 | 1.00 | 22.26 | 1.04 | 30.69 | 1.43 | 22.38 | 1.04 | 25.24 | 1.17 | 21.28 | 0.99 | 22.98 | 1.07 | 23.51 | 1.09 | 27.92 | 1.30 | 22.17 | 1.03 | 28.06 | 1.31 | 21.50 |
| US Births | 1192.20 | 1.03 | 586.93 | 0.51 | 399.00 | 0.35 | 419.73 | 0.36 | 526.33 | 0.46 | 574.93 | 0.50 | 441.70 | 0.38 | 557.87 | 0.48 | 424.93 | 0.37 | 422.00 | 0.37 | 504.40 | 0.44 | 452.87 | 0.39 | 1152.67 |
| GEOMEAN | 375.20 | 1.03 | 338.38 | 0.93 | 276.70 | 0.76 | 318.52 | 0.87 | 327.75 | 0.90 | 286.48 | 0.78 | 277.29 | 0.76 | 270.47 | 0.74 | 277.10 | 0.76 | 285.82 | 0.78 | 273.30 | 0.75 | 281.16 | 0.77 | 365.08 |

## C.3 ABLATION STUDIES

### C.3.1 MODULE DESIGN

We conduct module-wise ablation studies under the zero-shot setting. Specifically, **w/o SA** denotes the removal of the entire Scale-aware Alignment module, **w/o CM** indicates the exclusion of Continuous Masking, and **w/o RM** refers to the removal of Random Masking. The results, summarized in Table 8, demonstrate that the proposed Hybrid Masking strategy provides a robust training mechanism for encoder-only models, where the combination of the two masking approaches yields consistent performance improvements. Moreover, the incorporation of Scale-aware Alignment offers additional gains. A closer inspection reveals an interesting trend: Random Masking is more effective for short-term forecasting (prediction lengths of 96 and 192), while Continuous Masking contributes more to long-term forecasting (lengths of 336 and 720). In real-world scenarios where long-term forecasting is not required, adopting only Random Masking may serve as a more aggressive and efficient choice for maximizing model performance.

Table 8: Ablation results under the zero-shot setting. "w/o SA" denotes the removal of the entire Scale-aware Alignment module, "w/o CM" indicates the exclusion of Continuous Masking, and "w/o RM" refers to the removal of Random Masking.

| Models | | SATS | | w/o SA | | w/o CM | | w/o RM | |
|---|---|---|---|---|---|---|---|---|---|
| Metrics | | MSE | MAE | MSE | MAE | MSE | MAE | MSE | MAE |
| ETTh1 | 96 | 0.360 | 0.387 | 0.362 | 0.389 | 0.383 | 0.395 | 0.380 | 0.392 |
| | 192 | 0.395 | 0.409 | 0.398 | 0.411 | 0.427 | 0.422 | 0.414 | 0.412 |
| | 336 | 0.413 | 0.422 | 0.413 | 0.423 | 0.450 | 0.440 | 0.427 | 0.423 |
| | 720 | 0.413 | 0.438 | 0.414 | 0.441 | 0.487 | 0.485 | 0.415 | 0.436 |
| | AVG | 0.395 | 0.414 | 0.397 | 0.416 | 0.437 | 0.435 | 0.409 | 0.416 |
| ETTh2 | 96 | 0.273 | 0.331 | 0.275 | 0.337 | 0.283 | 0.327 | 0.279 | 0.328 |
| | 192 | 0.330 | 0.372 | 0.334 | 0.380 | 0.351 | 0.370 | 0.342 | 0.373 |
| | 336 | 0.353 | 0.396 | 0.361 | 0.403 | 0.375 | 0.394 | 0.375 | 0.400 |
| | 720 | 0.380 | 0.409 | 0.404 | 0.442 | 0.425 | 0.440 | 0.415 | 0.435 |
| | AVG | 0.334 | 0.377 | 0.343 | 0.391 | 0.358 | 0.383 | 0.353 | 0.384 |
| ETTm1 | 96 | 0.323 | 0.345 | 0.319 | 0.345 | 0.303 | 0.338 | 0.320 | 0.346 |
| | 192 | 0.352 | 0.364 | 0.346 | 0.366 | 0.333 | 0.361 | 0.353 | 0.368 |
| | 336 | 0.371 | 0.379 | 0.368 | 0.382 | 0.359 | 0.378 | 0.370 | 0.382 |
| | 720 | 0.401 | 0.403 | 0.404 | 0.403 | 0.411 | 0.409 | 0.400 | 0.405 |
| | AVG | 0.362 | 0.373 | 0.359 | 0.374 | 0.351 | 0.372 | 0.361 | 0.375 |
| ETTm2 | 96 | 0.167 | 0.251 | 0.185 | 0.258 | 0.187 | 0.266 | 0.203 | 0.277 |
| | 192 | 0.222 | 0.290 | 0.235 | 0.295 | 0.244 | 0.306 | 0.257 | 0.313 |
| | 336 | 0.269 | 0.323 | 0.280 | 0.326 | 0.297 | 0.340 | 0.306 | 0.346 |
| | 720 | 0.343 | 0.374 | 0.351 | 0.374 | 0.399 | 0.403 | 0.377 | 0.393 |
| | AVG | 0.250 | 0.309 | 0.263 | 0.313 | 0.282 | 0.329 | 0.286 | 0.332 |
| Weather | 96 | 0.162 | 0.217 | 0.172 | 0.220 | 0.172 | 0.220 | 0.180 | 0.212 |
| | 192 | 0.210 | 0.265 | 0.218 | 0.265 | 0.226 | 0.271 | 0.224 | 0.253 |
| | 336 | 0.258 | 0.302 | 0.262 | 0.297 | 0.275 | 0.309 | 0.269 | 0.286 |
| | 720 | 0.325 | 0.349 | 0.323 | 0.341 | 0.362 | 0.373 | 0.333 | 0.327 |
| | AVG | 0.239 | 0.283 | 0.244 | 0.281 | 0.259 | 0.293 | 0.251 | 0.269 |
| Average | | 0.316 | 0.351 | 0.321 | 0.355 | 0.338 | 0.362 | 0.332 | 0.355 |

### C.3.2 ALIGNMENT MECHANISM

In Table 9, we provide more detailed experimental results to further investigate the mechanism behind Scale-aware Alignment. This module is designed to simultaneously minimize the distance between mean embeddings and maximize the distance between maximal embeddings, thereby promoting alignment while mitigating feature collapse. To assess the impact of pooling strategies involved in this design, we conduct a series of ablation studies. Removing the repulsion component between maximal embeddings leads to a notable degradation in performance, which aligns with expectations due to the collapse of representation diversity. Additionally, substituting max pooling with min pooling or random pooling when defining the push-away objective consistently impairs performance, corroborating the intuition that maximal values encode the most informative features. Lastly, applying the alignment constraint solely via minimal distance between maximal embeddings proves insufficient, yielding results close to those without any alignment objective.

Table 9: Ablation study under the zero-shot setting. "w/o far" denotes the complete removal of the push-away (far) objective. "w minFar" uses the embedding derived from min pooling as the push-away target. "w randomFar" adopts a randomly pooled embedding as the push-away target. "w maxClose" replaces the push-away objective with a pull-close (near) objective, where the embedding is obtained via max pooling.

| Models | | SATS | | w/o far | | w minFar | | w randomFar | | w maxClose | |
|---|---|---|---|---|---|---|---|---|---|---|---|
| Metrics | | MSE | MAE | MSE | MAE | MSE | MAE | MSE | MAE | MSE | MAE |
| ETTh1 | 96 | 0.360 | 0.387 | 0.386 | 0.406 | 0.394 | 0.405 | 0.363 | 0.390 | 0.370 | 0.395 |
| | 192 | 0.395 | 0.409 | 0.411 | 0.418 | 0.422 | 0.423 | 0.401 | 0.412 | 0.403 | 0.414 |
| | 336 | 0.413 | 0.422 | 0.418 | 0.424 | 0.425 | 0.430 | 0.418 | 0.423 | 0.413 | 0.423 |
| | 720 | 0.413 | 0.438 | 0.422 | 0.443 | 0.421 | 0.446 | 0.419 | 0.440 | 0.410 | 0.435 |
| | AVG | 0.395 | 0.414 | 0.409 | 0.423 | 0.415 | 0.426 | 0.400 | 0.416 | 0.399 | 0.417 |
| ETTh2 | 96 | 0.273 | 0.331 | 0.288 | 0.346 | 0.283 | 0.348 | 0.273 | 0.336 | 0.271 | 0.332 |
| | 192 | 0.330 | 0.372 | 0.341 | 0.380 | 0.343 | 0.389 | 0.333 | 0.379 | 0.328 | 0.373 |
| | 336 | 0.353 | 0.396 | 0.388 | 0.412 | 0.375 | 0.415 | 0.366 | 0.406 | 0.357 | 0.397 |
| | 720 | 0.380 | 0.409 | 0.449 | 0.452 | 0.425 | 0.458 | 0.405 | 0.438 | 0.391 | 0.415 |
| | AVG | 0.334 | 0.377 | 0.367 | 0.397 | 0.357 | 0.402 | 0.344 | 0.390 | 0.337 | 0.379 |
| ETTm1 | 96 | 0.323 | 0.345 | 0.401 | 0.386 | 0.370 | 0.368 | 0.315 | 0.345 | 0.345 | 0.356 |
| | 192 | 0.352 | 0.364 | 0.372 | 0.376 | 0.381 | 0.382 | 0.347 | 0.367 | 0.367 | 0.373 |
| | 336 | 0.371 | 0.379 | 0.386 | 0.388 | 0.393 | 0.394 | 0.367 | 0.383 | 0.382 | 0.387 |
| | 720 | 0.401 | 0.403 | 0.414 | 0.410 | 0.418 | 0.415 | 0.396 | 0.406 | 0.411 | 0.409 |
| | AVG | 0.362 | 0.373 | 0.393 | 0.390 | 0.390 | 0.390 | 0.356 | 0.375 | 0.376 | 0.381 |
| ETTm2 | 96 | 0.167 | 0.251 | 0.262 | 0.291 | 0.183 | 0.269 | 0.185 | 0.272 | 0.171 | 0.259 |
| | 192 | 0.222 | 0.290 | 0.307 | 0.322 | 0.237 | 0.309 | 0.239 | 0.308 | 0.224 | 0.295 |
| | 336 | 0.269 | 0.323 | 0.304 | 0.362 | 0.296 | 0.348 | 0.288 | 0.341 | 0.272 | 0.327 |
| | 720 | 0.343 | 0.374 | 0.377 | 0.408 | 0.381 | 0.400 | 0.361 | 0.390 | 0.352 | 0.380 |
| | AVG | 0.250 | 0.309 | 0.312 | 0.345 | 0.274 | 0.332 | 0.268 | 0.328 | 0.255 | 0.315 |
| Weather | 96 | 0.162 | 0.217 | 0.206 | 0.250 | 0.179 | 0.231 | 0.170 | 0.224 | 0.168 | 0.220 |
| | 192 | 0.210 | 0.265 | 0.241 | 0.277 | 0.223 | 0.272 | 0.219 | 0.273 | 0.213 | 0.261 |
| | 336 | 0.258 | 0.302 | 0.281 | 0.306 | 0.266 | 0.303 | 0.262 | 0.306 | 0.257 | 0.294 |
| | 720 | 0.325 | 0.349 | 0.349 | 0.345 | 0.324 | 0.343 | 0.320 | 0.348 | 0.312 | 0.333 |
| | AVG | 0.239 | 0.283 | 0.269 | 0.294 | 0.248 | 0.287 | 0.243 | 0.288 | 0.237 | 0.277 |
| Average | | 0.316 | 0.351 | 0.350 | 0.370 | 0.337 | 0.367 | 0.322 | 0.359 | 0.321 | 0.354 |

## D STATEMENT ON THE USE OF LARGE LANGUAGE MODELS

In this work, Large Language Models were used solely to assist or polish the writing to improve clarity and presentation, and did not participate in any research design or literature review.

