# OpenReview forum: "Scale-Aware Pretraining of Time Series Foundation Models via Multi-Patch Token Alignment and Hybrid Masking"
_ICLR.cc/2026/Conference — ICLR 2026 Conference Withdrawn Submission_

### Official Review · Reviewer_StKk · 2025-10-29

**Soundness:** 2
**Presentation:** 2
**Contribution:** 2
**Rating:** 2
**Confidence:** 4

**Summary:**

In this study, the authors present a novel tokenisation procedure to address varying sampling frequencies in heterogeneous time series pre-training corpora. In particular, they adapt approaches with frequency-specific projection layers, such as MOIRAI [1], with (1) a scale-aware alignment to align the distinct projection layers and (2) a hybrid masking strategy to enforce the learning of meaningful time series features. Experiments are conducted using standardised forecasting benchmarks. An ablation study is conducted to investigate the importance of the proposed components.

___
[1] Woo, G. et al. "Unified training of universal time series forecasting transformers." ICML (2024).

**Strengths:**

1) The paper is well structured.
2) The authors make valuable efforts to establish a measure for the information density in time series data.
3) The authors evaluate their method on established forecasting benchmarks.
4) The authors conduct ablation studies to provide insights on the effectiveness of the proposed components.

**Weaknesses:**

Motivation:

1) The authors motivate their study with the limitations of previous works in the field of time series tokenisation: (1) 'fragmented token spaces' (see l. 79) in the case of dataset-specific patching and (2) 'information bottlenecks' (see l. 82) in domain-agnostic patching. However, these limitations are not grounded on examples from related works, making them appear artificial.

Related Works:

2) The authors do not discuss the most recent works in the field of time series tokenisation. For instance, works on domain-specific tokenisation [1] and wavelet-based tokenisation [2] should be included to provide a representative overview of the current literature.

Terminology:

3) The term 'time series foundation model' (as used in the title and throughout the manuscript) is misleading, as the proposed model is limited to forecasting tasks. A more suitable term in this context would be 'foundational forecasting model', since time series foundation model implies the capability to perform any arbitrary task in time series analysis, including classification and regression.

Methodology:

4) The authors propose scale-aware alignment of the projection layers using contrsative learning. While aligning the frequency-specific projections layers through pulling of the respective embeddings is somewhat intuitive, the pushing of max embeddings requires more ellaboration. This should prevent the frequency-specific projections to collapse, however, the rational behind max pooling within this context is not clear.
5) The hybrid masking strategy proposed in Section 3.1 is very similar to masking strategies proposed in previous works. For instance, in [1] the authors propose a dual masking strategy that combines random masking and post-fix masking (masking the second half of the time series). As post-fix masking is a special case of the contiguous masking, a comparison between hybrid and dual masking would be interesting: is post-fix masking sufficient or does contiguous masking further improve the quality of the learned features?

Experiments:

6) The model configurations presented in Table 1 resemble the MOIRAI [1] configurations (see Table 4 in [1]). While MOIRAI_B contains 91M parameters, SATS$_\text{B}$ with the same configurations contains only 70M parameters, which needs further elaboration.
7) The in-distribution forecasting results in Section 4.3 and the ablation studies in Section 4.4. are not reported across multiple seeds to ensure robustness.
8) The authors ablate the proposed components scale-aware alignment and hybrid masking, including random masking and contiduous masking, in Table 3. Although their work emphasises the importance of aligning frequency-specific projection layers (with the term scale-aware even appearing in the title and model name), the results presented in the second row of Table 3 suggest otherwise. Specifically, the proposed scale-aware alignment yields only marginal improvements of 1.56% MSE and 1.12% MAE.
9) The authors use contrastive learning to align the feature spaces of the frequency-specific projection layers. In Figure 4, however, they show that the feature spaces are not aligned, i.e. do not overlap, which is counterintuitive.
10) In Figure 4, the authors provide t-SNE plots, which are known to be sensitive to parameter tuning. Therefore, a latent space analysis using principal component analysis would be the fairer approach.
11) The authors introduce a model efficiency measure defined as 'the inverse of the product between the zero-shot error and the logarithm of model size' (see ll. 464-465). According to this definition, a naive predictor such as a simple seasonal trend model would achieve the highest efficiency score. This raises concerns about the validity and practical usefulness of the proposed efficiency measure.

Reproducibility:

12) The authors do not support reproducibility by making their code publicly available for evaluation.

Discussion:

13) The discussion of the experiments is too limited. The experiments section only provides numbers without their interpretation, while the conclusion section only provides a brief summary of the abstract.
14) The authors do not discuss the limitations of their work.
15) The authors do not discuss topics relevant for future work in the main manuscript.

___
[1] Turgut, Ö. et al. "Towards generalisable time series understanding across domains." arXiv preprint arXiv:2410.07299 (2024).

[2] Masserano, L. et al. "Enhancing Foundation Models for Time Series Forecasting via Wavelet-based Tokenization." ICML (2025).

[3] Woo, G. et al. "Unified training of universal time series forecasting transformers." ICML (2024).

**Questions:**

1) What is meant by 'fragmented token spaces' (see l. 79)?
2) What is meant by 'information bottlenecks' (see l. 82)?
3) The authors claim that using frequency-specific projection layers results in fragmented token spaces and propose contrastive learning to align these spaces. This raises the question why to use frequency-specific projection layers in the first place? Why not address this problem by using frequency-agnostic, i.e. universal, projection layers as current approaches [1]?
4) What is meant by 'Any-Variate Bias' (see l. 207)?

___
[1] Liu, X. et al. "Moirai-MoE: Empowering Time Series Foundation Models with Sparse Mixture of Experts." ICML (2025).

---

### Official Review · Reviewer_bkro · 2025-10-31

**Soundness:** 2
**Presentation:** 3
**Contribution:** 2
**Rating:** 4
**Confidence:** 4

**Summary:**

The paper tackles a core obstacle in pretraining TSFMs: token spaces fragment across datasets with different sampling rates. It proposes Scale-Aware preTraining for TSFMs (SATS), which treats each patch size as a distinct “scale,” assigns a small projection MLP per scale, and introduces a scale-alignment objective. A hybrid masking strategy mixes random masking and contiguous block masking to capture both fine-grain reconstruction cues and long-range temporal structure. Experiments on zero-shot forecasting and in-distribution evaluation report consistent gains over strong TSFM baselines.

**Strengths:**

1. This paper is well-motivated and easy to understand.
2. Modeling patch size explicitly as scale and aligning multi-scale token spaces via a parameter-free loss is conceptually crisp and easy to integrate into existing TSFMs.
3. The hybrid masking schedule balances local detail recovery and long-horizon dependencies, improving robustness across heterogeneous datasets.

**Weaknesses:**

1. The zero-shot evaluation is underpowered: among the five datasets, four are ETT variants that are highly homogeneous, so results may not convincingly demonstrate cross-domain generalization, weakening the strength of the claims. I encourage the authors to consider adopting a more diverse suite such as GIFT-Eval.
2. Although lighter than per-dataset heads, the approach still needs per-scale projection MLPs and patch-size choices, leaving some hyperparameter sensitivity issues.
3. Evaluation focuses on point forecasting (MSE/MAE). Generality to probabilistic metrics (e.g., CRPS) is not explored.

**Questions:**

1. How should practitioners choose the number of scales and patch lengths in a new domain?
2. Can the hybrid masking ratio and block-length distribution be scheduled or learned as a function of dataset frequency or training progress to reduce tuning?

---

### Official Review · Reviewer_oNMY · 2025-11-01

**Soundness:** 2
**Presentation:** 3
**Contribution:** 1
**Rating:** 2
**Confidence:** 4

**Summary:**

The paper takes Moirai as a baseline and attempts to solve the problem of aligning representations of different patch sizes in time series foundation models. It does so by introducing a contrastive loss, pulling mean embeddings and pushing max embeddings. It also uses a different masking strategy for pre-training, using random and contiguous block masking. It evaluates on the "LSTF" datasets for zero-shot setting and Monash TSF benchmark for in-distribution setting.

**Strengths:**

The paper is well presented in a logical manner, most details to understand the method are present, diagrams and equations are clear.

**Weaknesses:**

The experimentation setup is quite weak. There are many more recent TSFM baselines to consider, such as Toto, Chronos-Bolt, TimesFM, TiRex, etc. The datasets being used are also quite outdated, for TSFMs, the standard these days is to compare on more robust evals, such as GIFT-eval, or even newer benchmarks like fev-bench.

The ablations of scale-aware alignment is not quite convincing, the improvement in metrics is likely way too small to be considered significant. Also, it is unclear why mean and max embeddings are chosen for the contrastive loss in such a manner, apart from experimental results.

**Questions:**

Moirai uses a linear patch projection layer, while the paper uses an MLP, but there is no further details about the exact architecture of the MLP being used. Please provide more details on this, and it would also be good to see an ablation for this.

---

### Official Review · Reviewer_Kf6W · 2025-11-03

**Soundness:** 3
**Presentation:** 3
**Contribution:** 2
**Rating:** 6
**Confidence:** 3

**Summary:**

The authors present SATS, a Scale-Aware foundation model for Time Series that
aims to address the issue of fragmented token spaces and misaligned
representations in time series pretraining. The authors introduce a scale-aware
alignment mechanism in addition to a hybrid masking strategy that combines
random and contiguous masking to capture temporal dependencies at multiple
resolutions.

**Strengths:**

S1: The paper presents an interesting foundation model for time series data.

**Weaknesses:**

W1: In the experimental evaluation of SATS, one aspect that is not very clear
to me is why the authors did not compare it to the Moirai-MoE (Liu et al.,
2024b) model. The Moirai-MOE model is referenced in the paper, and it performs
better than the original Moirai model. Therefore, it would be beneficial if the
authors would compare SATS against it.

**Questions:**

Please see W1 under Weaknesses.

---

### Note · Authors · 2025-11-18

I have read and agree with the venue's withdrawal policy on behalf of myself and my co-authors.